# Spatially Distributed Characterization of Soil-Moisture Dynamics Using Travel-Time Distributions

Falk Heße[1], Matthias Zink[1], Rohini Kumar[1], Luis Samaniego[1], and Sabine Attinger[1,2]

[1]Department of Computational Hydrosystems, UFZ – Helmholtz Centre for Environmental Research, Leipzig, Germany.
[2]Institute of Geosciences, Faculty for Chemistry and Earth Sciences, Friedrich-Schiller-University Jena, Jena, Germany.

*Correspondence to:* Falk Heße(falk.hesse@ufz.de)

**Abstract.**

Travel-time distributions are a comprehensive tool for the characterization of hydrological system dynamics. Unlike streamflow hydrographs, they describe the movement and storage of water inside and through the hydrological system. Until recently, studies using such travel-time distributions have generally either been applied to simple (artificial toy) models or to real-world catchments using available time series, e.g., stable isotopes. Whereas the former are limited in their realism, the latter are limited in their use of available data sets. In our study, we employ a middle ground by using the mesoscale Hydrological Model (mHM) and apply it to a catchment in Central Germany. Being able to draw on multiple large data sets for calibration and verification, we generate a large array of spatially distributed states and fluxes. These hydrological outputs are then used to compute the travel-time distributions for every grid cell in the modeling domain. A statistical analysis shows the general soundness of the upscaling scheme employed in mHM and reveal precipitation, saturated soil moisture and potential evapotranspiration as important predictors for explaining the spatial heterogeneity of mean travel times. In addition, we demonstrate and discuss the high information content of mean travel times for characterization of internal hydrological processes.

**Keywords**

Travel-time distributions,mesoscale Hydrological Model (mHM), multiscale modeling, catchment hydrology, stochastic hydrology, model calibration

## 1   Introduction

The description of storage and transport of both water and dissolved contaminants in catchments is a challenging subject due to the high heterogeneity of the subsurface properties that govern their fate (*Dagan*, 1989). This heterogeneity, combined with a limited knowledge about the subsurface, results in high degrees of uncertainty. As a result, stochastic methods are often applied, where the relevant processes are modeled as being random (*Dagan*, 1986; *Rubin*, 2003). Amongst these methods, a powerful tool is the use of travel-time distributions (TTD's), where storage and transport inside the catchment are modeled from a Lagrangian perspective (*Rinaldo and Marani*, 1987; *Rinaldo et al.*, 1989). This means that the catchment itself or mean-

ingful parts of it is treated as a control volume (CV). The spatially complex array of different flow paths inside such a CV is consequently ignored and only inlet and outlet fluxes are used for the analysis (*Botter et al.*, 2010; *Rinaldo et al.*, 2011; *Botter*, 2012). This observation-based description of catchment dynamics makes TTD's a very robust tool. Although the application of TTD's goes back many decades (*Danckwerts*, 1953; *Niemi*, 1977), recent developments have strongly improved their theo-

retical foundations turning them into a versatile and coherent tool to characterize catchment dynamics (*Bertuzzo et al.*, 2013; *Benettin et al.*, 2015a; *Rinaldo et al.*, 2015; *Porporato and Calabrese*, 2015). Owing to this progress, *McMillan et al.* (2012) and *McDonnell and Beven* (2014) have opined that TTD's should be used routinely for hydrological model calibration, a notion that has been picked up with tremendous speed (*Windhorst et al.*, 2014; *Vereecken et al.*, 2015; *McGuire and McDonnell*, 2015). Independently but somewhat parallel to that, *Kitanidis* (2015) has recently pointed out, that the key to subsurface character-

ization is to use all available information. From this information-centered perspective, using TTD's have several advantages. First, the travel-time behavior is controlled by different factors than the hydrograph response. Whereas the latter is relating rainfall–runoff events the former is relating rainfall–runoff water (*McDonnell and Beven*, 2014; *Birkel and Soulsby*, 2015). Second, spatially distributed tracer experiments may dramatically increase the information content available for catchment characterization (*Birkel and Soulsby*, 2015).

This range of advantages have lead to a steady increase in both applied and theoretical studies using TTD's for the description of catchment dynamics. Applied studies here means that data from real-world sites are used (*McGuire et al.*, 2005; *Cardenas*, 2007; *Broxton et al.*, 2009; *Tetzlaff et al.*, 2011; *Dunn et al.*, 2012; *Hrachowitz et al.*, 2013, 2015; *Harman*, 2015). Here the advantage is that the data used for the analysis do not suffer from model errors or other conceptual limitations. However, such data are generally limited in amount (e.g., tracer or isotope time series limited to a few years only, although

*Hrachowitz et al.* (2009) used time series of up to 17 years) and variety (only a limited number of data types are available). As a result, such studies might fail to find long-term trends, establish connections between travel-time behavior and specific catchment properties or to investigate the impact of certain hydraulic regimes that are only rarely occurring (e.g., drought or extremely rainy months). In the second category, we find theoretical studies, that either use a very simplified computational model to focus on specific questions (*Rinaldo et al.*, 2006; *Duffy*, 2010; *Botter et al.*, 2010; *van der Velde et al.*, 2012;

*Benettin et al.*, 2015a; *Porporato and Calabrese*, 2015) or employ more realistic hydrological models that provide a large data set typically not available in real-world sites *Sayama and McDonnell* (2009); *Fenicia et al.* (2010); *McMillan et al.* (2012). Such theoretical studies allow a more thorough and detailed analysis of the involved processes and their interdependence but may suffer from an oversimplified model setup for in- and outflux generation.

Our study falls into the latter category such that we use a hydrological model, i.e., the mesoscale Hydrological Model

(mHM) (*Samaniego et al.*, 2010a; *Kumar et al.*, 2013a), to generate the fluxes and states for the analysis. Using detailed data of precipitation, land cover, morphology and soil type as inputs, mHM is able to provide continuous simulations of spatially distributed fluxes (e.g., groundwater recharge or evapotranspiration) and states (e.g., soil moisture) as outputs. By employing mHM, which is a spatially-distributed hydrological model, we are, however, able to extend these prior studies to a spatially-distributed travel-time analysis. This makes it possible to address several types of investigation. First, it allows

for a comprehensive description of the flow and transport dynamics taking place in the catchment. The spatial distribution

of such dynamics can then be related to e.g., land cover and physical properties of the soil as well as to driving forces like precipitation to determine dominant predictive factors. In addition, it allows to investigate how certain parametrizations of the mHM model are related to the travel-time behavior of the catchment. This opens the way for a more robust model calibration of hydrological models using additional datasets (*McDonnell and Beven*, 2014; *Birkel and Soulsby*, 2015; *Kitanidis*, 2015).

As a case study, we use a ca. 1000 $km^2$ catchment in Central Germany for which detailed morphological and climatological data are available to parametrize mHM. In addition, the chosen catchment is the location of the Hainich Critical Zone Exploratory, a comprehensive monitoring network used within the Collaborative Research Center AquaDiva (*Küsel et al.*, 2016). AquaDiva seeks to elucidate the critical role of water fluxes connecting surface conditions with biogeochemical functions in the subsurface. One of the goals of this project is to understand how far signal of surface properties, like land cover or land management, can be traced into the subsurface water and solute dynamics. Spatially explicit travel-time distributions are the perfect analytical tool to investigate such questions.

By virtue of using the modeled data from mHM, we are able to address several questions that have not been investigated before. First, how are spatially-distributed quantities, in particular land-cover, precipitation and soil type, impacting travel-time behavior in the soil? Unlike earlier model-based studies, mHM is a spatially-distributed hydrological model. We can therefore add to prior knowledge by investigating the travel-time behavior for every mHM grid cell and relate it to its geophysical and climatic properties. Next, how do different hydrological regimes (wet vs dry) impact travel-time behavior in the soil? Here we investigate the impact of changing external conditions (meteorological factors) using the long time-series of modeled fluxes and states. Finally, what is the inter-connection between travel-time behavior and specific conceptualization of different hydrological processes, and how may these connections be used for further improvement of model parameterization? Investigating the impact of model-specific conceptualizations on the predicted travel-time behavior can provide a better understanding on how actual measurement may be connected to certain model parameters. For the quantitative analysis, we focus on soil moisture, only, i.e., we exclude groundwater. This was necessary, due to the implementation of groundwater in mHM as a linear reservoir. Although variations, i.e., fluxes, of the groundwater level can be represented well (*Rakovec et al.*, 2016) the total amount remains elusive. This is a common feature of hydrological models (*Fan*, 2015) and mHM is no exception. Furthermore, we consider this restriction to be acceptable within the scope of our study, i.e., elucidation of the spatio-temporal dynamics of TTD's. Groundwater by definition is far less impacted by the spatial distribution of precipitation or land cover. In addition, *Benettin et al.* (2015b) recently showed that TTD's show little temporal variability compared to soil moisture.

To present our results on such questions, the rest of the paper is organized as follows: In Section 2 we describe the numerical and analytical tools used in this study. Thus comprises the framework of travel-time distributions, as applied in this study, as well as the relevant features of mHM. In Section 3, we present the results of our study and demonstrate how they relate to the questions raised above. Finally, in Section 4 we summarize our main findings in light these questions and draw some conclusions.

## 2   Methods

In the following, we provide a short overview of the analytical and numerical tools and methods used in this study. We start by introducing the concept of travel-time distributions. In the following, we use the nomenclature as given by *Benettin et al.* (2015a) and the theoretical framework by *Botter et al.* (2010). In addition to that, we give a short overview of the numerical model (mHM) which was used for the calculation of the states and fluxes. Finally, we introduce the catchment used in our study.

### 2.1   Travel-time distributions for a single control volume

Travel-time distributions are a stochastic description of the dynamic of a water parcel moving through a given control volume (CV). The definition of such a control volume for real-world situation is often arbitrary to some extent (see e.g., the schematic in Figure 1). Within the context of this study, we used a spatially distributed model where the catchment is partitioned in regular grid cells (for more details see Section 2.2 below). Consequently, the boundaries of our CV were given by the grid cells of the model.

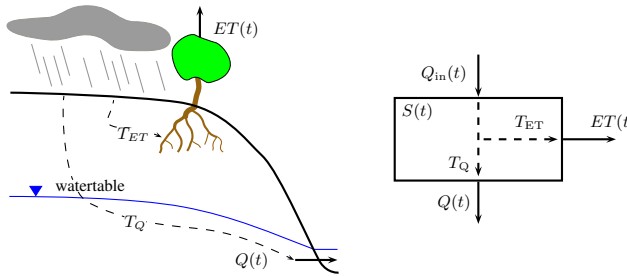

**Figure 1.** Water movement inside a hill slope (physical schematic on the left and conceptual schematic on the right).

Given that such a CV can be reasonably defined, it is clear that the dynamics of a water parcel is determined by the in- and out-fluxes, that are changing the water content inside it. The time evolution of the water content $S$ inside such a CV is then given by the following balance equation

$$\frac{d}{dt}S(t) = Q_{\text{in}}(t) - Q_{\text{out}}(t) = J(t) - (ET(t) + Q(t)). \tag{1}$$

Equation (1) is a simple initial-value problem with the in-flux $Q_{\text{in}}(t)$ given by the effective precipitation $J(t)$ whereas the out-flux $Q_{\text{out}}(t)$ is given by evapotranspiration $ET(t)$ and runoff per grid cell $Q(t)$.

To denote the different times involved in the dynamic of a water parcel, we followed the notation of *Benettin et al.* (2015a). Chronological time was accordingly denoted with $t$, whereas the water parcel entered the CV at $t_{\text{in}}$ and left at $t_{\text{ex}}$. At any given time $t'$ in between these two points, any water parcel can therefore be characterized by two different properties; its age $T_A$ as well as its (remaining) life expectancy $T_E$ (see Figure 2).

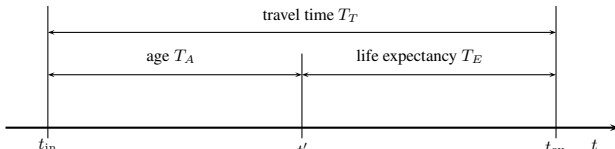

**Figure 2.** Schematic of different times associated with the travel-time dynamics of a water parcel. Age $T_A$ is the time elapsed at $t'$ since the water parcel entered the CV at $t_{in}$, whereas life expectancy $T_E$ is the time remaining at $t'$ until the water parcel leaves at $t_{ex}$.

In their paper, *Benettin et al.* (2015a) emphasize the two interpretations that originate from these two points of view. Age is a backwards concept referring to the time passed since the beginning. The associated travel time distribution is therefore called the backward TTD. The concept of backward TTD's is of particular interest for the characterization of e.g., a water sample, since its composition is determined by the age of the water in the CV. Life expectancy, on the other hand, is a forward
concept since it is referring to the time still left until exit from the CV. The associated travel time distribution is therefore called the forward TTD. Such forward TTD's are relevant e.g., for tracer test, since the concentration of an ideal tracer at the outlet is given by the TTD of its associated water parcel.

In order to derive the TTD's associated with the forward and backward formulation, *Botter et al.* (2011) presented a derivation using only the states and fluxes inside the CV as well as what they call an age function (for more information on their
derivation we also refer to *Botter et al.* (2010) and the references therein). In the following, we assume a uniform age function only. This means that the age distribution of the water leaving the CV is the same as the age distribution of the water inside the CV, i.e., no age preference of the outflow generating processes (discharge and ET) exists. This decision became necessary, since we could not yet draw on any data for the age distribution of water at the outlet of the catchment. As a result, we were not able to compare the predictions of different age functions to any measurements and therefore determine the most ade-
quate description. In absence of such data, the most appropriate choice is the one involving the least amount of information, which is given by the assumption of uniform sampling. Using this assumption, we can state the following for the forward formulation

$$\overrightarrow{p}_Q(T_E, t_{in}) = \frac{Q(t)}{\theta(t_{in})S(t)} \exp\left(-\int_{T_E} \frac{Q(t') + ET(t')}{S(t')} dt'\right) \tag{2}$$

with $T_E = t - t_{in}, t > t_{in}$, i.e., the time from the moment the water parcel entered the reservoir until now. The function $\theta$
in Equation (2) is called the partition function (*Botter et al.*, 2010, 2011) and can be derived using the following formula

$$\theta(t_{in}) = \int_{t_{in}}^{\infty} \frac{Q(t)}{S(t)} \exp\left(-\int_{T_E} \frac{Q(t') + ET(t')}{S(t')} dt'\right) dt. \tag{3}$$

This partition function describes the portion of the water parcel, entering the CV at $t_{in}$, that is contributing eventually to discharge as opposed to evapotranspiration. It is consequently a dimensionless number between $0$ and $1$.

For the backward formulation, we can state the following

$$\overleftarrow{p}_Q(T_A, t_{\text{ex}}) = \frac{J(t)}{S(t)} \exp\left(-\int\limits_{T_A} \frac{Q(t') + ET(t')}{S(t')} dt'\right) \tag{4}$$

with $T_A = t_{\text{ex}} - t, t < t_{\text{ex}}$, i.e., the time from now until the moment the water parcel leaves the reservoir.

Both these formulations determine the travel time of the water leaving as discharge. The TTD's for the water leaving as
evapotranspiration can be determined in an analogous way.

## 2.2   Numerical model

We used a spatially distributed, grid-based mesoscale Hydrological Model (mHM; *Samaniego et al.* (2010a); *Kumar et al.* (2013a)) to generate the states and fluxes needed for the TDD analysis described above. The model uses the grid cell as a primary hydrological unit and models the following dominant hydrological process: interception, snow accumulation
and melting, root zone soil moisture dynamics, evapotranspiration, surface flow, interflows, recharge and baseflow. The total runoff generated at each grid cell is routed to the neighboring downstream cell following the river network using the Muskingum-Cunge routing algorithm. Interested reader may refer to (*Samaniego et al.*, 2010a) for further details on the model components. The model code is open source and can be downloaded from www.ufz.de/mhm. The model has been successfully applied to a number of river basins across Germany, USA and Europe (*Samaniego et al.*, 2010a, b; *Kumar et al.*, 2010,
2013a, b; *Samaniego et al.*, 2013; *Livneh et al.*, 2015; *Thober et al.*, 2015; *Rakovec et al.*, 2016).

An important and unique feature of mHM is its Multiscale Parameter Regionalization (MPR), that explicitly accounts for subgrid variability of basin physical characteristics such as terrain, soil, vegetation, and geological properties (*Samaniego et al.*, 2010a; *Kumar et al.*, 2013a). The model considers different levels of spatial resolution to better account for spatial heterogeneity of inputs, forcings and the modeled hydrological processes (see schematic in Figure 3). The smallest scale (called $l_0$ within
the mHM nomenclature) is representing morphological factors, like elevation, soil type, land cover etc. On the other hand, meteorological inputs can be represented on a larger scale (called $l_2$ within mHM). The modeling of the hydrology is done on a third scale (called $l_1$ within mHM) that can vary depending, e.g., on catchment size or computational resources. Based on the MPR technique, morphological inputs are linked to internal model parameters (e.g., through the use of pedo-transfer functions) and a set of regional coefficients (or global parameters, $\gamma$). In a second step, the internal parameters are upscaled to
the resolution of the hydrological processes, i.e., $l_1$, using parameter specific upscaling operators. Thus, MPR takes indirectly subgrid variabilities into account. The global parameters ($\gamma$) are space and time invariant and are inferred via a calibration procedure. mHM has 66 global parameters, which is a reasonable number for an optimization problem and is therefore able to avoid overparameterization. Further details on MPR can be found in *Samaniego et al.* (2010a); *Kumar et al.* (2013a).

Relevant to this study is near-surface and root-zone hydrological process, which are computed using different concep-
tualizations. In the upmost layer ($x_3$ in Figure 3) water content is estimated using the infiltration excess approach similar to the HBV model (*Bergström*, 1995), but enhanced to account for multiple layers. Within these layers, the water is either

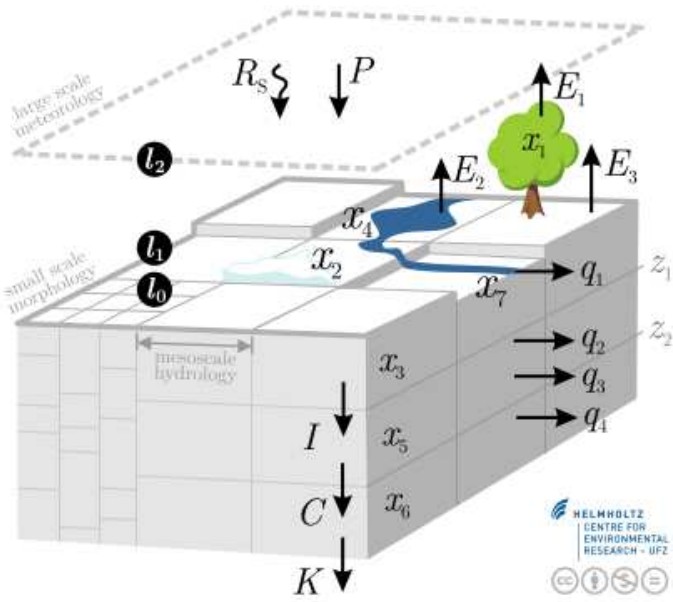

**Figure 3.** Schematic of the mesocale hydrological model used in the study, depicting the different scales as well as the states and fluxes represented in a single cell.

percolating into deeper layers or evapotranspirates to the atmosphere. Therefore, the root zone is characterized by effective parameters for porosity, saturated hydraulic conductivity and the permanent wilting point, which are estimated based on the pedotransfer functions of *Zacharias and Wessolek* (2007). These effective parameters are estimated due to transfer functions from the global parameters, which are determined during the calibration process. Evapotranspiration is estimated based on potential evapotranspiration, root water uptake and water availability in layer $x_3$. In the second layer ($x_5$ in Figure 3) two different types of interflow take place. Slow interflow $q_3$ is implemented using a power-law model, whereas fast interflow $q_2$ is triggered when a threshold value $\gamma_{TV}$ is reached, i.e

$$q_3 = \gamma_s x_5^{(1+\alpha)}, \tag{5a}$$

$$q_2 = \begin{cases} \gamma_f(x_5 - \gamma_{TV}), & \text{if } x_5 > \gamma_{TV} \\ 0, & \text{otherwise} \end{cases}. \tag{5b}$$

In the third level ($x_5$ in Figure 3) baseflow $q_4$ is generated using a simple reservoir model, i.e.,

$$q_4 = \gamma_b x_6. \tag{6}$$

These runoff generation processes are represented at every grid cell of mHM. The sum of direct runoff $q_1$ (not used for the analyis), interflows and baseflow constitutes the grid specific total runoff which is then routed through a river network. Interested readers may refer to *Samaniego et al.* (2010a) or www.ufz.de/mhm (user manual) for further details on mHM.

As motivated in the Introduction, we followed the concept of e.g., *Botter et al.* (2010) and *Benettin et al.* (2015b) and di-
vided the subsurface into two distinct zones; the soil zone (called root zone by *Botter et al.* (2010) and shallow storage by *Benettin et al.* (2015b)) and the saturated zone (called called groundwater region by *Botter et al.* (2010) and deep storage by *Benettin et al.* (2015b)). All analysis in our study was performed with respect to the former. This was seen necessary due to the large uncertainties associated with storage estimation of the deeper regions. Whereas mHM has been demonstrated to provide good estimates for soil moisture (*Rakovec et al.*, 2016), storage estimates for groundwater ($x_6$ in Figure 3) are far less
reliable. Focusing on the soil zone, only, was seen justified since the focus of our study was the investigation of spatially distributed factors like precipitation, land cover and soil type, which have comparably little impact on groundwater dynamics. Furthermore, *Benettin et al.* (2015b) demonstrated that travel-time behavior in the deeper zone has comparably little temporal variability, too.

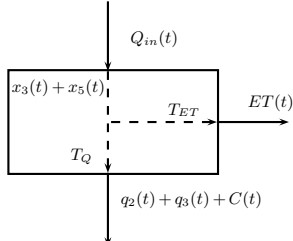

**Figure 4.** States and fluxes as computed by mHM (see Figure 3) for the derivation of TTD's using Equations 2 to (3) (see Figure 1).

For computation of the TTD's according to Equations (2) to (4), we used for the storage $S$ the combined estimates of layer
$x_3$ and $x_5$ of mHM (see Figures 3 and 4). For $ET$ we used fluxes the evapotranspiration from $x_3$ and for $Q$ we used $q_2$, $q_3$ and $C$. Conceptually the interflow is generated in the unsaturated zone (reservoirs $x_3$ and $x_5$) within mHM. Thus, using the interflow as the outflow from the unsaturated zone for deriving the travel times is a valid assumption. Our delineation of shallow and deeper storage was therefore more similar to *Benettin et al.* (2015b) than to *Botter et al.* (2010).

### 2.3 Study area and model set-up

In this study, we used a mesoscale catchment in Central Germany with a drainage area of approximately $1000 \ km^2$ to the gauging station at Nägelstedt (see Figure 5). The catchment is the headwaters of the Unstrut river basin, and was selected in this study for its relevance to the Collaborative Research Center AquaDiva (*Küsel et al.*, 2016). The terrain elevation within the catchment ranges between $170 \ m$ and $520 \ m$ with the higher regions in the west and south being the forested hill chain of the Hainich (see Figure 5). The forested area covers approximately $17\%$ of the catchment, while $78\%$ of the area is covered

by crop/grassland. The remaining 5% is urban/build up area. The area is characterized by continental climatic conditions with a mean annual precipitation of approximately 660 $mm$ and a mean temperature of approximately 8 $^\circ$C.

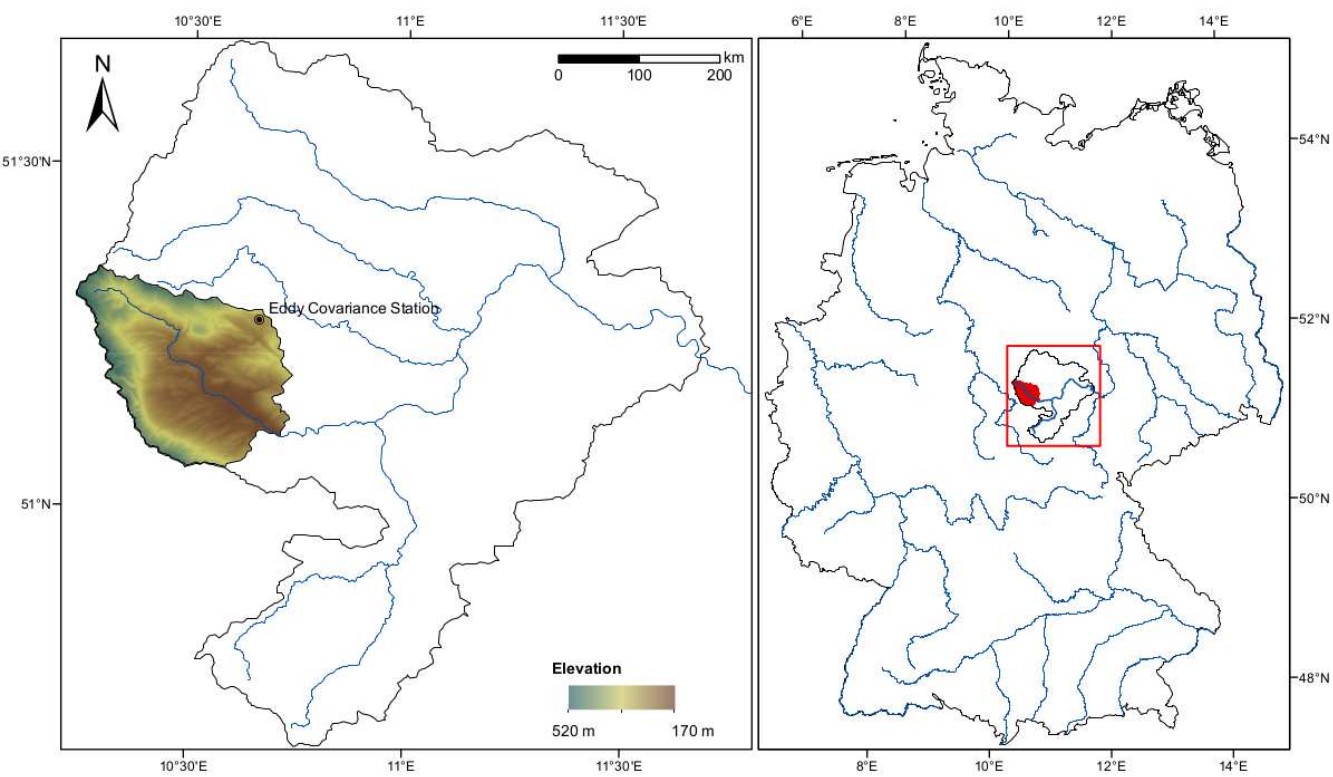

**Figure 5.** Left: Catchment (highlighted) used in the study shown within the larger confines of the Unstrut catchment (area enclosed by continuous line). The larger rivers of the catchment are shown in blue. The colorbar shows the elevation (in $m$) of the study catchment. Right: Unstrut catchment within the larger confines of Germany. The axis descriptions denote the latitude and longitude values.

We established mHM over the study catchment and performed numerical simulations on several resolutions ranging from 200 $m$ to 2 $km$. The model was forced using daily gridded fields of precipitation, air temperature and potential evapotran-
5    spiration. The point datasets for the precipitation and air temperature at several raingauges and weather stations located in and around the catchment were acquired from the German Meteorological Service (DWD). These point stations were then interpolated on regular grids using an external drift kriging interpolation procedure wherein the terrain elevation was used as an external drift (*Samaniego et al.*, 2013). The potential evapotranspiration was estimated using the *Hargreaves and Samani* (1985) method. Other datasets required to set-up the model include digital elevation model (DEM) and derived terrain prop-
10    erties like slope, aspect, flow direction, catchment boundary; soil and geological maps were provided by the Federal Institute for Geosciences and Natural Resources (BGR) and meta data such as sand and clay contents, bulk density, horizon depths, dominant hydrogeological classes; CORINE land cover information for the years 1990, 2000 and 2005 available from the

European Environment Agency (EEA); and runoff data for the catchment outlet provided by the European Water Agency (EWA) and the Global Runoff

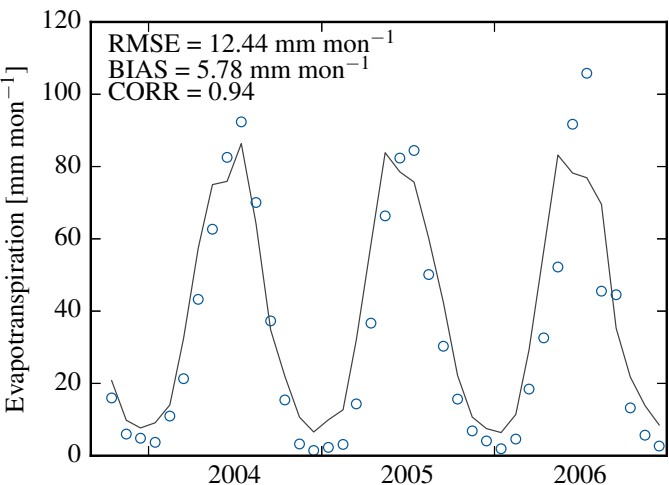

**Figure 6.** Comparison between monthly measured and modeled evapotranspiration (ET) at the eddy-covariance station Mehrsted (see Figure 5).

The model simulations were performed for the period 1950 to 2005. The first five years of the data were used to warm-up the model to acquire plausible initial conditions. We therefore discarded the first five years of simulations and the further analyses were performed using model outputs for the period $1955 - 2005$. The model showed quite good performance with $NSE > 0.8$ for the daily discharge simulations at the Nägelstedt station. Other statistics like bias and correlations were also within a satisfactory range. To further validate our model prediction, we used measurements from a single Eddy Covariance measurement station inside the study area (see Figure 5). This comparison also showed a good agreement between both measurements and model prediction (see Figure 6).

## 3   Results and discussion

In this section, we present and discuss the results which have been derived using the methods described above. We will begin in the following by demonstrating and exemplify our general research procedure by virtue of a singly yet representative example.

The starting point for the derivation of soil travel times were the states and fluxes as computed through mHM. Using the time series of soil moisture, evapotranspiration, interflow and recharge, we used Equation (2) to compute the travel time distribution (see Figure 7 a). One of the problems when computing forward TTD's by virtue of Equation (2) is that all the water entering the CV at time $t_{\text{in}}$ must leave until the end of the computing period. This means that a certain amount at the end of the available time series could not be used for the analysis. To determine this period, we computed $\theta_{\text{in}}$ with respect to discharge as well as to evapotranspiration. Adding up both values for a given $t_{\text{in}}$ should add up to one, i.e., all water

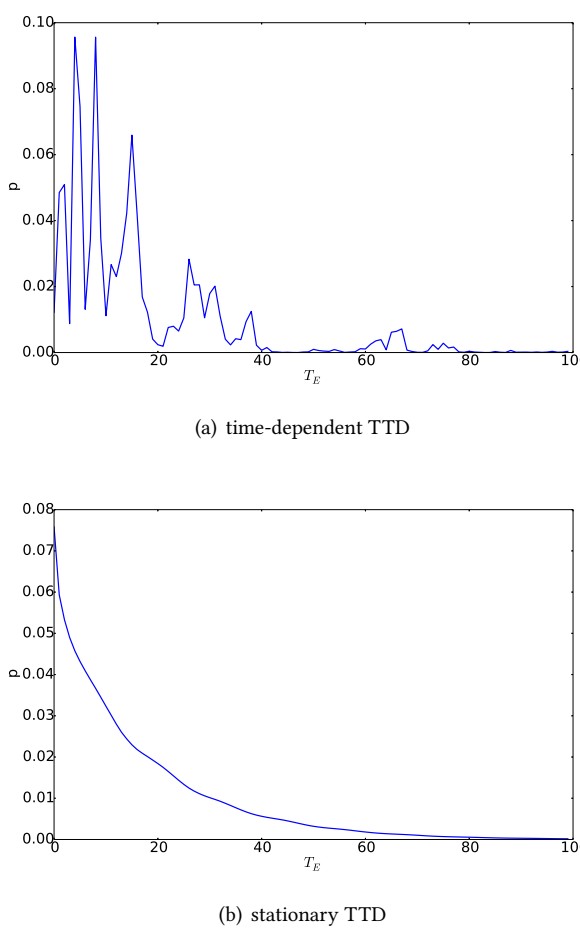

(a) time-dependent TTD

(b) stationary TTD

**Figure 7.** Forward TTD of soil moisture with respect to mean travel time (in months) for a single cell in the Nägelstedt catchment. Panel (a) shows the time-dependent TTD derived using Equation (2) for a given $t_{\mathrm{in}}$. Panel (b) shows the stationary TTD derived using Equation (7).

that entered at $t_{\mathrm{in}}$ has left within the available time frame. A value smaller that therefore indicates that some amount of the water is still inside the CV with possible error inducing effects on the calculation of the TTD's. Analyzing this behavior, we concluded that close to 2 years at the end of the available time series had to be excluded for the calculations of the TTD's (data not shown). The shape of the resulting time-dependent distributions varied strongly, depending in particular on rainfall events that triggered the mobilization of older water stored within the soil. Another factor, although not apparent from Figure 7, was the water content, i.e., the state of the soil itself. As been demonstrated by *Niemi* (1977), soil response to rain events is strongly different between wet or dry conditions.

To disentangle this event-driven as well as state-dependent effects from other factors that influence the water movement in the soil, we averaged these time dependent distributions. As a result, we got the stationary TTD's for every cell

$$\overrightarrow{p}_Q(T_E) = \int \frac{Q(t)}{\theta(t_{\text{in}})S(t)} \exp\left(-\int\limits_{T_E} \frac{Q(t') + ET(t')}{S(t')} dt'\right) dt_{\text{in}} \tag{7}$$

with $T_E = t - t_{\text{in}}, t > t_{\text{in}}$.

In all investigated cases, these stationary TTD's could be well approximated by an exponential-like behavior (see Figure 7 b). Behavior as seen for stationary TTD's is often assumed to be valid for TTD's in general and are consequently modeled using exponential or gamma distributions (*Małoszewski and Zuber*, 1982). Recent works however, have questioned this generalization by emphasizing the time-dependent nature of TTD's (*Duffy*, 2010; *Botter et al.*, 2011). The examples given in Figure 7 exemplify these concerns by illustrating their respective origins. Consequently, we acknowledged the inherent differences

between these two TTD's. Furthermore, the study area falls within a humid region with soils being generally wet and rainfall being evenly distributed throughout the year. Under these conditions the assumption of (quasi) stationary TDDs is reasonable (*Tetzlaff et al.*, 2007; *Hrachowitz et al.*, 2009). These stationary TTD's provided the basis for all following analysis, due to allowing the description of the average hydrological response of the catchment. In addition, we also focused on travel-time behavior under specific hydrological regimes, i.e., wet and dry conditions, providing a more detailed understanding of the

catchment.

For our statistical analysis, we used these stationary TTD's, which, due to their exponential-like behavior, can be characterized by its expected value $\tau$. We will call this value mean life expectancy (or mean age respectively) in the following. Estimating this value for every mHM cell provided a single measure for the travel-time behavior in the soil without the otherwise dominating impact of single precipitation events (see Figure 8). One feature that became immediately apparent

were the long travel times in urban areas (see Figure 8 a). This can be explained by the fact that these areas are largely sealed, resulting in low infiltration rates and consequently low turnover rates inside the soil. To disentangle this sealing effect from the soil behavior, we discarded cells inside urban regions from our analysis (see Figure 8 b). This allowed us to investigate the interplay between soil properties and travel-time behavior apart from such artificial influences.

### 3.1 Impact of modeling resolution

Due to its multiscale parameterization, mHM is able to model catchment dynamics at different spatial resolution with the same set of calibration parameters (see e.g., *Samaniego et al.* (2010a) or *Kumar et al.* (2013a)). Within the context of TTD's, this feature may be used to investigate the potential influence of age-dependent outflow generation. The mathematical theory for including such age dependency has been developed independently by different groups and recently been unified using the umbrella term of StorAge Selection (SAS) functions (*Rinaldo et al.*, 2015; *Harman*, 2015). These functions fully describe

the sampling behavior of the catchment with respect to the age distribution of the stored water when discharge is generated.

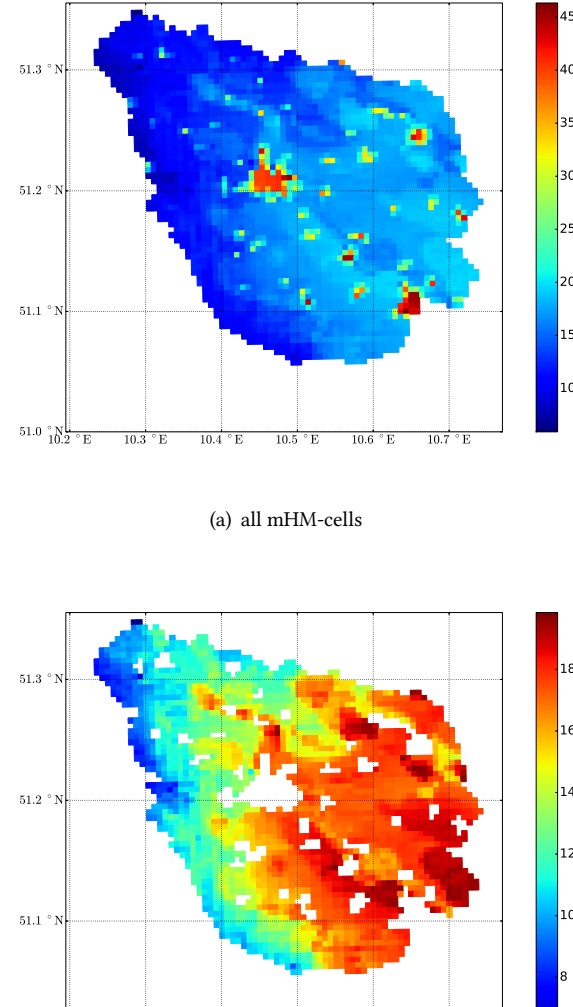

(a) all mHM-cells

(b) non-urban mHM-cells

**Figure 8.** Mean life expectancy (in months) of soil moisture derived by Equation 7 for the Nägelstedt catchment (see also Figure 1 for comparison) once for all mHM cells (left) and for all non-urban cells.

Discharge from a catchment may e.g., be primarily composed of younger or older water or it may show no preference to age whatsoever. SAS functions are therefore a concise mathematical representation of this behavior.

On a physical basis, such preference for different water age should be interpreted as the result of complex mixing processes taking place in the subsurface of the catchment (*Botter*, 2012; *Benettin et al.*, 2013; *van der Velde et al.*, 2012). In order

to determine the appropriate SAS function for a given catchment, predictions using different functions could be compared with measurements. Alternatively, the form of the SAS function can be determined by using a physically based catchment model (*Cornaton and Perrochet*, 2006a, b). As already mentioned above, we could not directly infer, which form of a SAS function would be the most appropriate choice for our catchment. Instead we calculated the mean life expectancy for our catchment on different scales using the uniform SAS function. We motivated this choice by the principle of least information (or principle of maximum entropy) stating that amongst different alternatives, the one with the least amount of information should be chosen. Without any additional constraints, a uniform distribution is usually associated with maximum ignorance, therefore, motivating the use of the uniform SAS function.

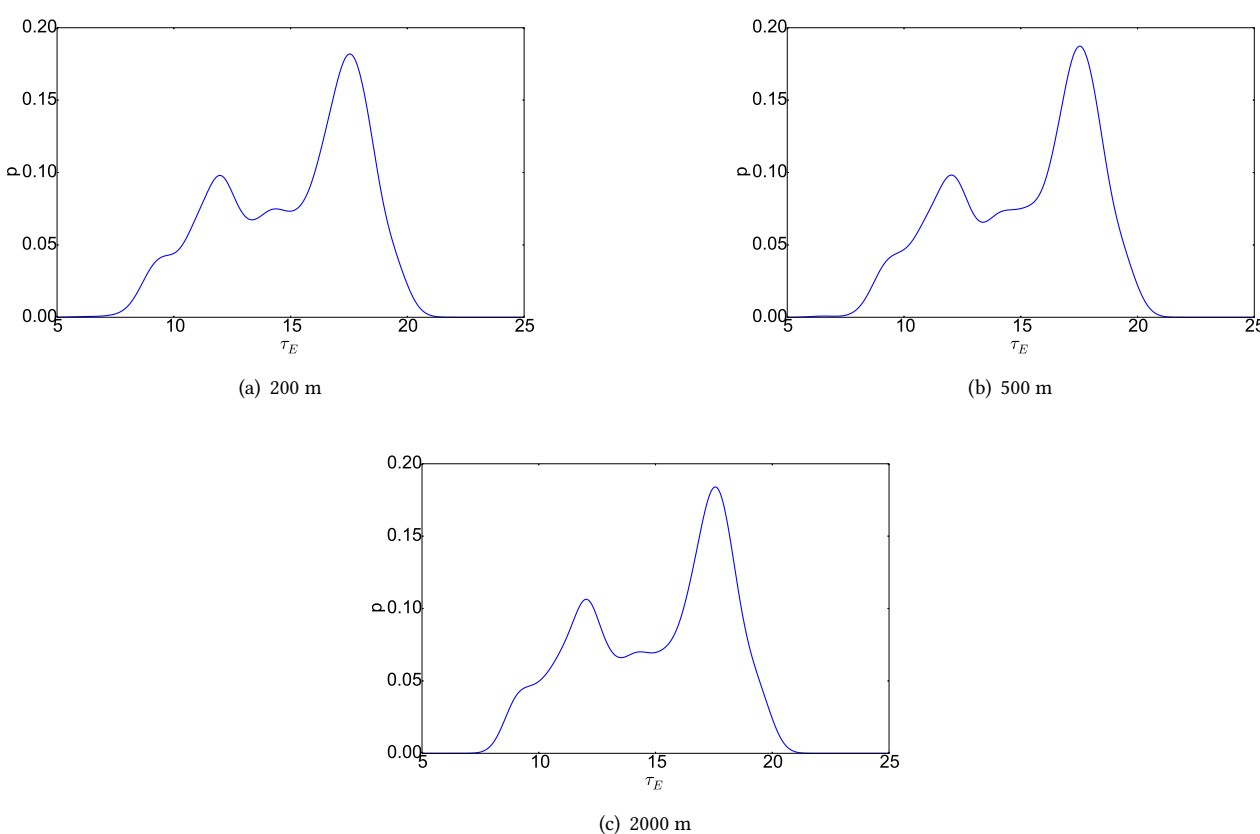

(a) 200 m

(b) 500 m

(c) 2000 m

**Figure 9.** Kernel density estimate of the mean life expectancy (in months) of soil moisture for several grid sizes in the Nägelstedt catchment.

To estimate the possible influence of this decision, we reasoned that a scale-dependent bias in the estimation of travel-time behavior would indicate the existence and possible strength of such an error. This is due to the multi-scale nature of mHM, where subgrid heterogeneity is taken into account by virtue of the Multiscale Parameter Regionalization. Using a smaller grid size would make this heterogeneity explicit and therefore reveal any possible unaccounted subgrid influence. Results from our simulations showed no discernible differences in the statistical distribution of mean life expectancy (see Figure 9). Using

a smaller resolution had positive effect on statistical estimation procedure due to the increase in data points. In addition, we saw more extreme values due to small scale features that were smeared out on coarser resolutions. Other than these two changes, we noted only minor changes in the statistics of mean life expectancy. We therefore concluded that, within the limits of the spatial scales tested here, mixing processes inside our catchment have no major impact on mean life expectancy.

5     We are aware, that this assessment is only covering one possible source of age-dependent outflow behavior and that other unresolved heterogeneity (at even smaller scales or due to other subsurface properties not accounted for in mHM) would influence the outflow generation as well. We therefore regard our conclusions as tentative and open to revision once actual measurements become available.

    However, our investigation gave us the ability to find a good trade-off between computational costs and data amount for 10   the following statistical analyses. We therefore used a data set from simulations using a grid size of $500\ m$.

## 3.2   Statistical analysis of mean life expectancy

The mean life expectancy $\tau$ of a water parcel inside a catchment is the result of a complex interplay of morphological and climatological factors. Several recent studies have therefore tried to determine their relative importance under varying conditions (*McGuire et al.*, 2005; *Cardenas*, 2007; *Broxton et al.*, 2009; *Tetzlaff et al.*, 2009, 2011). Contrary to these studies where field 15   measurements were used, we used results from computational simulations only. This gave us a much larger dataset, both in time and space, from which we could infer the relative impact of different factors, in particular meteorological (precipitation), land surface (land cover, leaf-are index) and subsurface (soil) properties. Notably, our approach differs from *Hrachowitz et al.* (2009) such that our analysis is based on model-derived gridded simulations of TDDs as compared to the observation-based basin-wise quantification of TDDs.

20     In the first step, we determined for every cell the statistical relationship between the mean life expectancy $\tau$ and a number of potential predictors like average precipitation, soil depth, soil type or leaf-area index (LAI). Similar to *Hrachowitz et al.* (2009), we used the coefficient of determination $R^2$ to quantify the strength of the statistical relationship. This quantity equals to one minus the ratio of the remaining variance vs. the total variance of the data themselves. It is therefore a measure of the variance explained by the model (which was always assumed to be linear in our study).

25   ### 3.2.1   Precipitation

The analysis above showed the strong impact of precipitation on the event-based TTD's (see Figure 7). It is therefore to be expected to exert strong control on the steady-state TTD's as well. In our model two different quantities can be distinguished: first, the precipitation itself as well as, second, the effective precipitation. The latter value is here defined as the water flux that is actually entering the soil, i.e., corrected by surface runoff (through sealing), canopy interception and snowmelt. While 30   the precipitation can be measured with high accuracy, it is the effective precipitation that directly impacts the soil-moisture dynamics.

    The scatterplot of both data sets against the mean life expectancy show a significant negative correlation between them (see Figure 10). This negative relationship can be explained such that precipitation events apply pressure to the water already

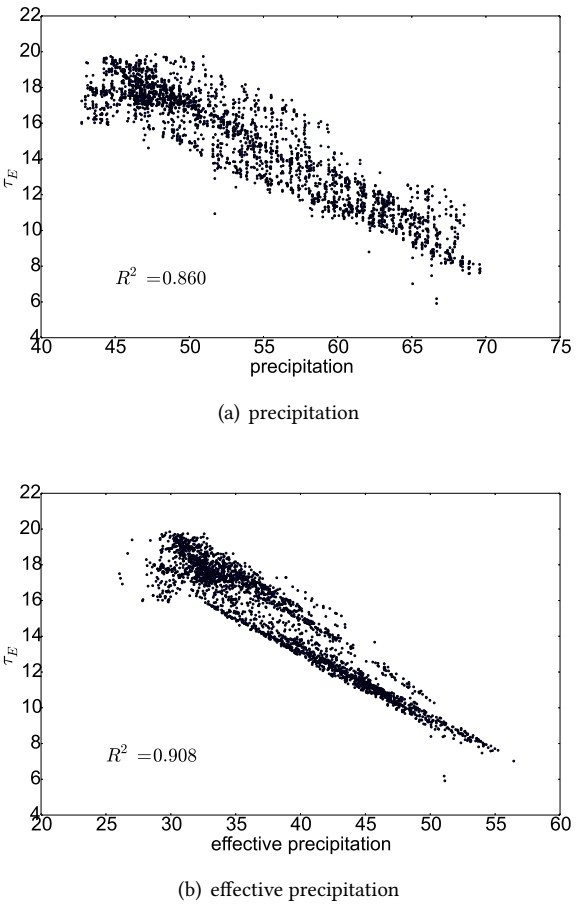

(a) precipitation

(b) effective precipitation

**Figure 10.** Scatterplot of mean life expectancy (in months) of soil moisture vs. monthly values (in $mm$) for precipitation (Panel (a)) and effective precipitation (Panel (b)).

stored in the soil. Instead of immediately traveling through the soil, the water from these events rather pushes older water out. Strong precipitation events therefore lead to a 'flushing out' of the soil and cause a shorter life expectancy.

### 3.2.2 Terrain elevation

In our next analysis, we used the physical elevation as a variable for our regression model. The height can simply be derived from the digital elevation model (DEM), which, in mHM, is represented using data obtained from the Shuttle Radar Topography Mission.

Using a scatter plot for visualizing the statistical relationship between mean life expectancy and the DEM showed a negative correlation (see Figure 11 a), i.e., longer life expectancy correlated with lower heights of the terrain, and with a linear coefficient of determination of $R^2 = 0.668$. Since no direct causal connection can be drawn between physical elevation and

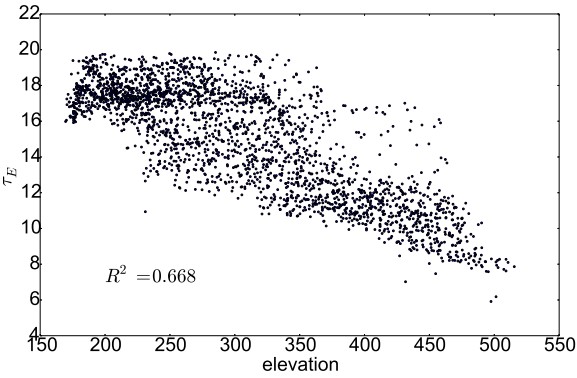

**Figure 11.** Scatterplot of mean life expectancy (in months) of soil moisture vs. elevation (in $m$).

travel-time behavior, such a high value is indicative of underlying mechanisms. One of these is the aforementioned precipitation, since higher altitudes are correlated with stronger mean precipitation levels (linear coefficient of determination of $R^2 = 0.812$). Performing a multiple linear regression, including precipitation and saturated soil moisture (discussed below), showed strong correlation between these variables (data not shown). It therefore stands to reason to attribute potential causal

effects to these covariates, only.

### 3.2.3    Evapotranspiration

Evapotranspiration is directly influencing the form of a TTD (see e.g., Equation 2). Consequently, we anticipated a strong correlation between mean evapotranspiration rates and mean life expectancy.

With respect to evapotranspiration, two different definitions are typically distinguished: potential evapotranspiration

(PET) and actual evapotranspiration (AET). As implied by its name, PET describes the maximum possible rate of evapotranspiration at a given site. This value is dependent on quantities like solar radiation and temperature that can generally be measured with good accuracy (*Samani*, 2000). Using theoretical models, good estimates can therefore be provided for PET at a given site (*Almorox et al.*, 2015). On the other side, AET is a real quantity that can be measured. In principle, in situ measurements can therefore provide good estimates (e.g., the eddy-covariance method). In practice, however, exact measurements

are hampered by a series of factors (*Wang and Dickinson*, 2012). As a consequence, PET can often be estimated with higher accuracy than AET.

Scatterplots of both PET and AET show a positive correlation between evapotranspiration and mean life expectancy in general (see Figure 12). This correlation is more pronounced for AET with a coefficient of determination of $R^2 = 0.496$ vs. only $R^2 = 0.259$ for PET.

Contrary to precipitation, which is an inflow mechanism, ET is an outflow mechanism. It is not pushing but rather sucking the water out of the CV, which explains the difference in behavior of precipitation and ET. The lower relative strength of the correlation (compared to precipitation) can be explained such that ET is only one of the two outflow mechanisms (the

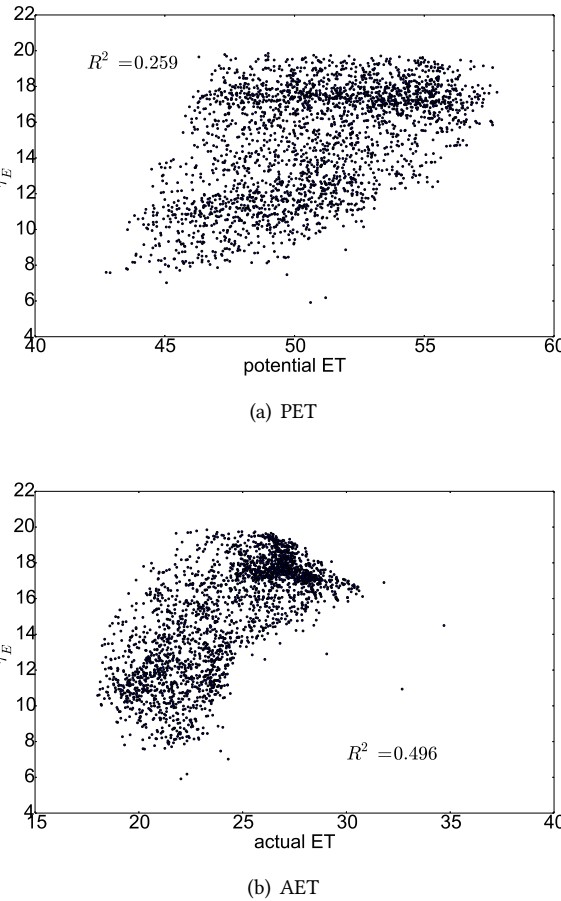

(a) PET

(b) AET

**Figure 12.** Scatterplot of mean life expectancy (in months) of soil moisture vs. monthly evapotranspiration values (in $mm$). Displayed are both potential evapotranspiration (Panel (a)) and actual evapotranspiration (Panel (b)).

other being discharge). The relative stronger impact of AET compared to PET was also anticipated. AET is directly used in Equation (2) for the calculation of TTD's, whereas PET is only coupled by virtue of an additional function.

As explained above, for real-world situations, better estimates can often be provided for PET. The higher explanatory power of AET has therefore to be balanced with its often less accurate estimate. Depending on the accuracy of measurements of AET, PET estimates may be a better predictor of mean life expectancy.

### 3.2.4 Land cover properties

Land cover is an important interface controlling the strength of incoming fluxes through artificial and natural sealing. In mHM, three different land-cover types are distinguished: forest, crop/grassland and urban area. As explained above, we excluded mHM cells inside urban area from our analysis in order to better focus on the soil properties themselves. To further

elucidate possible influence of the remaining land cover types, we separated the catchment into forest and crop/grassland and calculated the mean travel times separately.

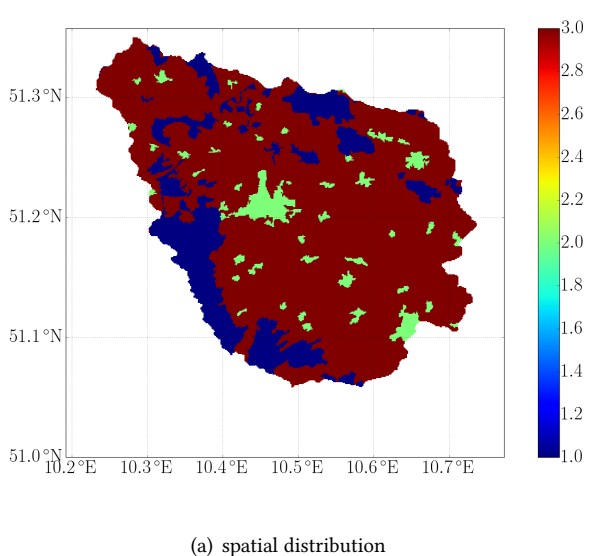

(a) spatial distribution

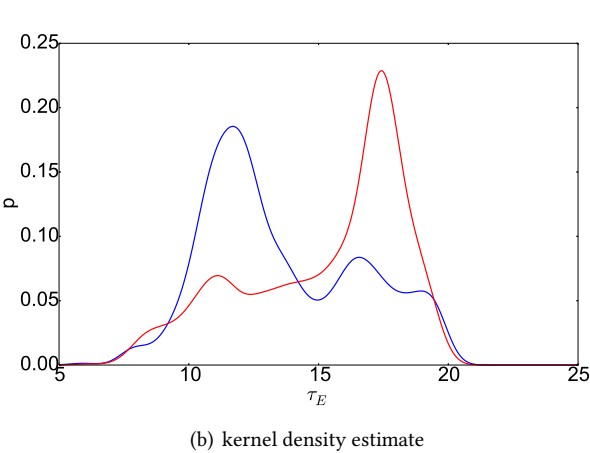

(b) kernel density estimate

**Figure 13.** Land cover in the Nägelstedt catchment (blue = forest, green = urban, red = crop/grassland). Panel (a) shows the spatial distribution of land cover in the highest resolution $l_0$ and Panel (b) shows the kernel density estimates of the mean life expectancy (in months) of soil moisture for the land cover types.

Estimating the PDF of the mean life expectancy for both land cover types separately, revealed strong differences between them both in shape of the respective PDF and the range of values (see Figure 13). As shown above, results for the combined data set showed a distinct bimodal behavior (see Figure 9). In contrast to that, the PDF for both land cover types were almost

unimodal. The most dominant peaks of every singly PDF coincided with the two peaks of the combined PDF. The behavior of the latter can therefore – to some degree – be considered to be a superposition of the former.

The relationship between these two land cover types was such that forest resulted in much shorter mean travel times compared to crop/grassland. This pronounced difference may be partially due to a correlation with precipitation patters that have already been shown to exert a strong influence on travel-time behavior. Forest in the study catchment (as well as in Germany in general) is disproportionately found in hilly and mountainous regions. These regions in turn show stronger precipitation values. The tendency depicted in Figure 13 may therefore be also caused by this covariate. However, this correlation between forested and high-precipitation area would not explain the distinct differences between both land-cover types. Another factor, overlapping with the former, may be due to the differences in water uptake. Trees are rooted into deeper soil layers compared to crop and grass and are therefore able to access a larger part of the subsurface water body. This larger access combined with the higher precipitation values as well as other factors would explain the almost non-overlapping travel-time behavior demonstrated in Figure 13.

In addition to this classification scheme, mHM uses the leaf area index (LAI) to describe land cover properties. The LAI describes the ratio of the cell that is effectively covered by plant canopy. Due to the already established influence on evapotranspiration (see above), it stands to reason to expect an influence on the mean life expectancy as well. Comparing LAI class and land cover reveals a strong overlap between both (see Figure 13 (a) and Figure 14 (a)). Roughly, forest land cover corresponds with LAI class $1 - 4$, urban land cover corresponds with LAI class $5$ and grassland corresponds with LAI class $6 - 10$.

Using the same approach as above, i.e., investigating the mean life expectancy for every LAI class independently, consequently revealed the same overall tendency for LAI classes compared to land cover types (data not shown). This was anticipated due to the aforementioned overlap between the two classification schemes. In addition, we saw little diversity for LAI classes within the same land cover class (data not shown).

However, this tendency was not present when using the actual leaf-area values associated with every LAI class. These values could be constant over the year e.g., in case of coniferous forest or vary strongly e.g., in case of deciduous forest. To make values from different LAI classes comparable, we averaged the respective values year-wise. A scatter plot of leaf area index vs. mean life expectancy does not show any strong correlation between the two with similar ranges of values being found for almost all LAI values (see Figure 14 (b)). This discrepancy can be explained by the implementation of the LAI in mHM. In contrast to the land cover type, that is used for the determination of ET processes in the upmost soil layer, LAI values are only used for interception and do consequently not directly influence travel-time behavior. As a result, any possible relationship between LAI and TTD's is therefore biased and conclusions from our results must take into account this limitation critically.

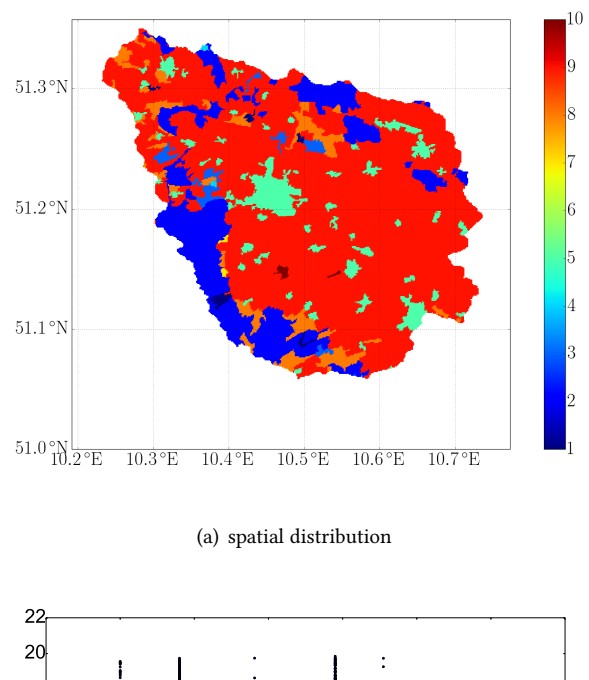

(a) spatial distribution

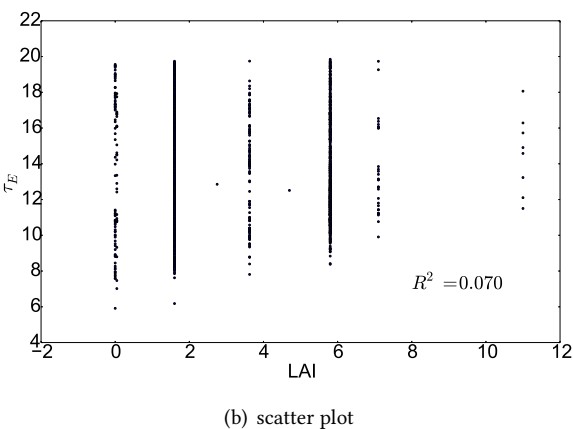

(b) scatter plot

**Figure 14.** Leaf area index (LAI) in the Nägelstedt catchment. Panel (a) shows the spatial distribution and Panel (b) shows the scatter plot of mean life expectancy (in months) of soil moisture vs. LAI.

### 3.2.5 Soil properties

An important input parameter in mHM is the soil type inside every cell. This property is implemented in mHM using the German soil data base Bodenübersichtskarte 1:1.000.000 (BÜK 1000) (*Federal Institute for Geosciences and Natural Resources (BGR)*, 1998).

5     Due to this relevance in the model, we anticipated a strong impact of the soil type in a cell on the resulting mean life expectancy. Estimating the PDF of mean travel times for every soil type individually, did indeed show significant differences between them (see Figure 15). Soil classes found in the geographically lower regions of the catchment generally show longer

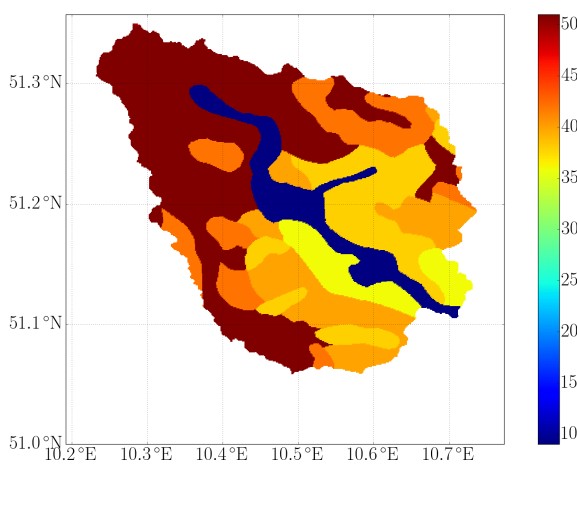

(a) spatial distribution

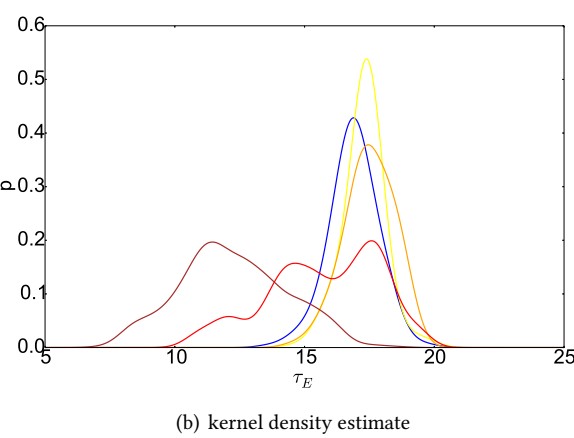

(b) kernel density estimate

**Figure 15.** Soil classes in the Nägelstedt catchment. Panel (a) shows the spatial distribution and Panel (b) shows the kernel density estimate of the mean life expectancy (in months) of soil moisture for selected soil classes. Blue curve represents soil class 9 (36% sand and 10% clay), yellow curve represents soil class 38 (12% sand and 15% clay), orange curve represents soil class 40 (10% sand and 19% clay), red curve represents soil class 42 (7% sand and 39% clay) and brown curve represents soil class 51 (19% sand and 70% clay).

mean travel times with a unimodal distribution shape, whereas soil types in the geographically higher regions correspond with generally shorter mean travel times with the shape of the distributions being less regular. This qualitative analysis reveals some overlap with the land cover distributions as well as mean precipitation rates. It is consequently not possible to directly infer causal correlation from statistical correlation.

In addition, the soil class is a symbolic variable, i.e., its values only indicate a certain type of soil but does not directly relate to any numerical quantity associated with this soil type. Consequently, we could not infer any quantitative connection between soil types and resulting travel-time behavior.

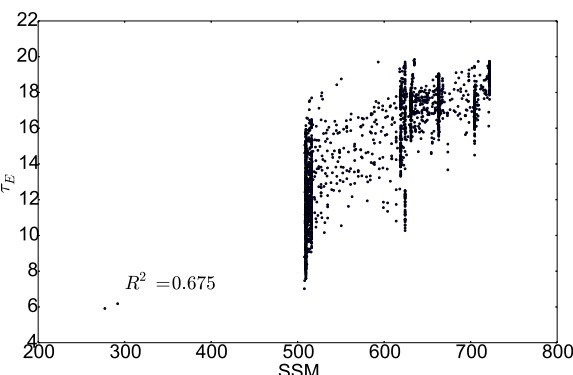

**Figure 16.** Scatter plot of mean life expectancy (in months) of soil moisture vs. saturated soil moisture (in $mm$).

To address this problem, we used the saturated soil moisture of the soil. This quantity is the amount of pore space per cell that can be potentially filled with water (porosity times the depth of root-zone soil layer). Its value is determined in mHM through pedo-transfer functions using the soil textural information on percentage of sand, clay and bulk density. Comparing these values in every single cell with the mean life expectancy shows a very strong statistical relationship with a coefficient of determination $R^2 = 0.675$.

The high correlation values of the saturated soil moisture can be explained by a mixture of causal and statistical factors. On one hand, it is reasonable to expect the total amount of storage to be filled with water to have a significant effect on the resulting travel-time bahvior. On the other hand, the soil tapes show a strong overlap with other factors like precipitation levels and land cove types that have already been discussed above.

### 3.3 Statistical analysis of mean age

As described above, the difference between the forward and backward formulations of travel time has long been acknowl-edged (*Niemi*, 1977) and many studies have investigated their relationship (*Cornaton and Perrochet*, 2006a; *Botter*, 2012; *Benettin et al.*, 2013; *Harman*, 2015; *Benettin et al.*, 2015a). Both these formulations are linked by virtue of the so called Niemi relation

$$J(t_{\text{in}})\theta(t_{\text{in}})\overrightarrow{p}(t - t_{\text{in}}|t_{\text{in}}) = Q(t)\overleftarrow{p}(t - t_{\text{in}}|t), \tag{8}$$

which can be derived by considering a water parcel entering the CV at $t_{\text{in}}$ and leaving at $t$. Consequently, mean life expectancy and age only coincide in case of steady-state conditions. As a result, we also investigated the behavior of mean age to elucidate connections and differences between forward and backward formulations for our catchment.

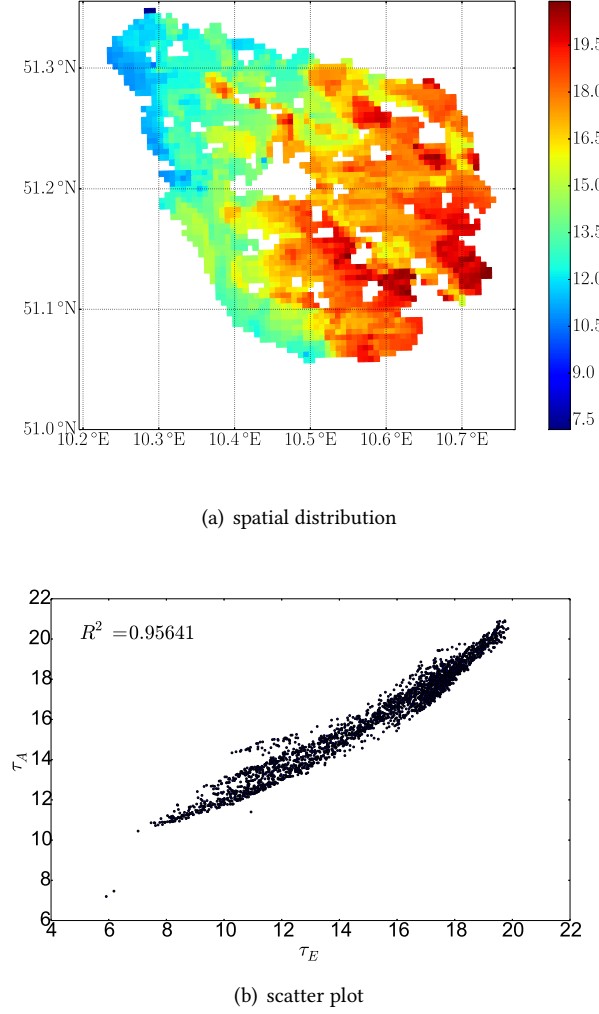

(a) spatial distribution

(b) scatter plot

**Figure 17.** Mean age of soil moisture in the Nägelstedt catchment. Panel (a) shows spatial distribution and Panel (b) shows scatter plot of mean age vs. mean life expectancy (see Equation 8) in months.

Visually comparing mean age (see Figure 17) and mean life expectancy (see Figure 8 b) in the Nägelstedt catchment showed

5   strong qualitative and quantitative similarities. Accordingly, we also got a very strong statistical relationship between these two quantities with a coefficient of determination of $R^2 = 0.956$. Overall, the relationship was very linear with mean age values falling short of mean life expectancy for both small and large values.

|  | Pre | PET | SSM |
| --- | --- | --- | --- |
| mean life expectancy | 0.860 | 0.260 | 0.675 |
| mean age | 0.728 | 0.143 | 0.711 |

**Table 1.** $R^2$ values for several predictors of mean travel time

Due to the mathematical and physical similarities, such a strong connection was anticipated. To further investigate possible origins of their respective differences, we performed the same statistical analysis for mean age.

To that end, we considered proxy variables that have already been shown to have a considerable impact on travel-time behavior. As demonstrated by the analysis above, these were precipitation (Pre), potential evapotranspiration (PET) and saturated soil moisture (SSM) as proxies for influx, outflux and state respectively. Results showed overall the same trend for mean age and life expectancy with respect to these predictors (see Table 1). Precipitation was the most dominant factor for both quantities with the saturated soil moisture being a close second. This is in contrast to e.g., *Benettin et al.* (2015a), who emphasized the role of the outfluxes for the time evolution of both age and life expectancy. In our analysis, we saw that proxy variables for influx and state showed strongest correlations with mean travel-time behavior. On the other hand, PET, which is a good proxy for one of the two outfluxes, showed only moderately strong correlations with said behavior. In case of mean age, this relationship was even weaker compared to the other two (precipitation and saturated soil moisture). Since we could not provide a proxy variable for the other outflux, i.e., discharge, we excluded this quantity from our analysis.

### 3.4 Joint impact of multiple variables on mean travel times

In the analysis above, the statistical relationship between mean travel-time behavior and a number of variables was presented and discussed. This was done for every variable individually to elucidate its possible impact on mean travel times. In addition to this simple analysis, we also investigated the joint impact of several variables. Such results can be of relevance for prediction, i.e., using a set of variables to predict travel times in a given CV.

To that end, we used the variables that had been shown to have the highest impact individually, i.e., precipitation, saturated soil moisture and potential evapotranspiration, and performed a multiple linear regression. Simple linear regression had already demonstrated that both precipitation and saturated soil moisture could explain a significant amount of the variability contained in the dataset. Combining these factors could therefore improve the predictability even further. We therefore applied Forward Stepwise Selection to generate a series of models with increasing complexity. The fist single-variable model consequently used only precipitation as the variable with the highest single $R^2$ value. Next, the double-variable model used both precipitation and saturated soil moisture and the most complex three-variable model used precipitation, saturated soil moisture and potential evapotranspiration jointly.

Results for the default case, showed that, compared to using only one variable (precipitation), using two variables for the regression (precipitation and saturated soil moisture) improved the predictability of mean travel times (see Table 2). This was expected since both variables alone provided already high $R^2$ values. In addition, precipitation and saturated soil moisture did

| | Pre | Pre + SSM | Pre + SSM + PET |
|---|---|---|---|
| mean life expectancy | 0.860 | 0.911 | 0.913 |

**Table 2.** $R^2$ values of for several regression models of increasing complexity.

only show moderate correlation ($R^2 = 0.451$), so adding the latter variable added new information to the prediction model. The correlation that existed between precipitation and saturated soil moisture is explained by the an orographic effect, i.e., hilly regions in the catchment, with typically lower values of saturated soil moisture, also show higher precipitation values. In contrast, using three variables (precipitation, saturated soil moisture and potential evapotranspiration) resulted in almost negligible improvement (see Table 2). This is due to the already lower impact of PET compared to precipitation and saturated soil moisture. In addition, PET showed comparably stronger correlation with both precipitation and saturated soil moisture (data not shown), therefore adding only little new information compared to the other two variables. Such low impact of outgoing fluxes compared to precipitation has already been reported before, for the case of synthetic toy models (*Daly and Porporato*, 2006). Moreover our results agree with the findings of *Hrachowitz et al.* (2009), who also reported similarly strong explanatory power of climatic variables like precipitation as well as soil and land surface properties.

### 3.5 Impact of hydrological regime on travel-time behavior

The analysis above revealed the strong impact of the influx (i.e., precipitation) as well as the state variable (i.e., saturated soil moisture) on the travel-time behavior. To further elucidate their impact, we investigated travel-time behavior independently for different hydrological regimes during the considered period of time, i.e., from 1955 - 2005. To that end we partitioned the available time series into regimes based on soil moisture (state variable) and precipitation events (influx).

In the first case, we averaged the time series of mean saturated soil moisture in the whole Nägelstedt catchment for every year, i.e., 50 years in total. Next, we divided the resulting time series such that years with an average soil moisture content above 85th percentile of the time series were labeled as wet years. In contrast, years with an average soil moisture content below 15th percentile of the time series were labels as dry years. This annual partitioning was seen necessary due to the strong annual fluctuations of this variable. Finally, we performed the same analysis as describe above for both – now smaller – datasets.

Using results from dry years only (see Figure 18), showed a similar qualitative travel-time behavior but strong quantitative contrast compared to the mean travel-time behavior discussed above (see Figure 8). Compared to the general case, mean life expectancy was much larger in dry years. In addition, dry years exhibit a wider range of possible values with the largest one (over 50 months) being almost 4 times a large as the smallest one (approximately 12 months).

Wet years on the other hand, exhibit a very small range with the smallest value (approximately 5 months) being roughly only half as large as the largest value (approximately 11 months) (see Figure 19). Compared to the general case, where the largest value (approximately 20 months) were roughly 3 times as large as the smallest value (approximately 6 months), these

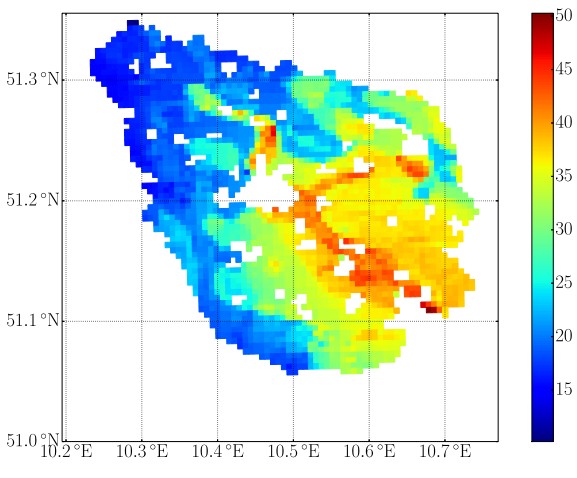

(a) spatial distribution

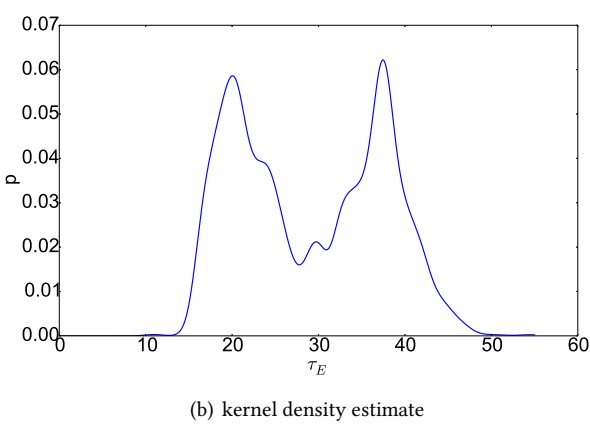

(b) kernel density estimate

**Figure 18.** Mean life expectancy of soi moisture in the Nägelstedt catchment in dry years. Panel (a) shows the spatial distribution and Panel (b) shows the kernel density of mean life expectancy (in months).

two scenarios fall on either side of this spectrum. This stark discrepancy demonstrates again the strong impact of the state variable (soil moisture) on travel-time behavior. Another difference between the mean travel-time behavior in wet years and the general case is the unimodal distribution of the former. The analysis above revealed how the bimodal behavior is mostly due to the different soil types and therefore reflects the strong impact on this property on the overall soil-moisture dynamics. The disappearance of this bimodal behavior is therefore reflective of how the soil becomes 'forced into line' when being filled up with water leveling prior differences.

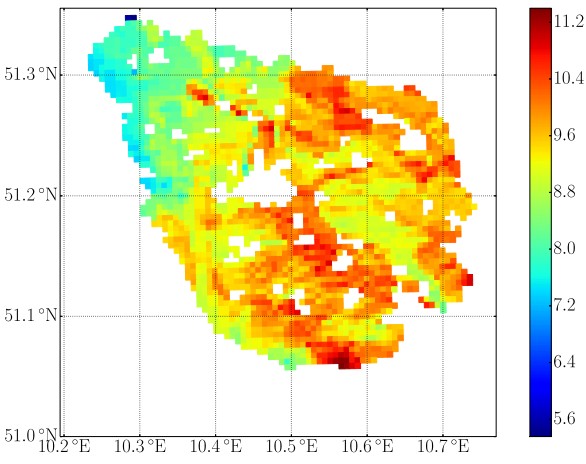

(a) spatial distribution

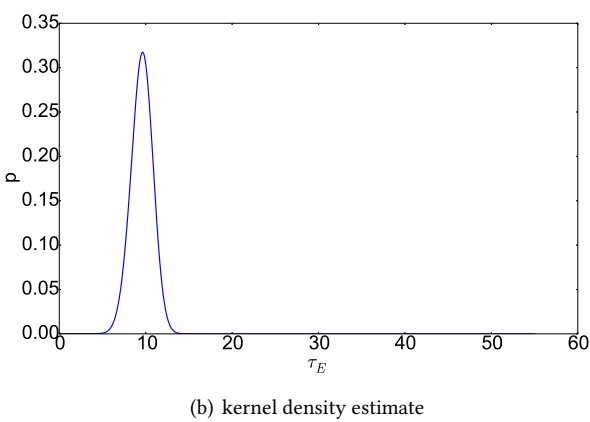

(b) kernel density estimate

**Figure 19.** Mean life expectancy of soi moisture in the Nägelstedt catchment in wet years. Panel (a) shows the spatial distribution and Panel (b) shows the kernel density of mean life expectancy (in months).

|  | Pre | PET | SSM |
|---|---|---|---|
| wet years | 0.374 | 0.084 | 0.388 |
| dry years | 0.781 | 0.223 | 0.834 |

**Table 3.** $R^2$ values for several predictors of mean travel time (as caused by wet and dry years).

In addition, results showed different statistical dependency of travel-time behavior with respect to precipitation, PET and SSM (see Table 3). Dry years showed very similar correlation values compared to the general case (see Table 1 a). On the other side, correlation values for wet years were remarkably smaller.

In the second case, we also investigated travel-time behavior depending influx, i.e., in case of months having above average precipitation values (rainy months). To that end, we constrained our analysis to forward travel-time distributions which were triggered by heavy rain events. This means that, in analogy to the analysis above we only used months with precipitation values above the 97th percentile and performed again the same analysis for the reduced dataset.

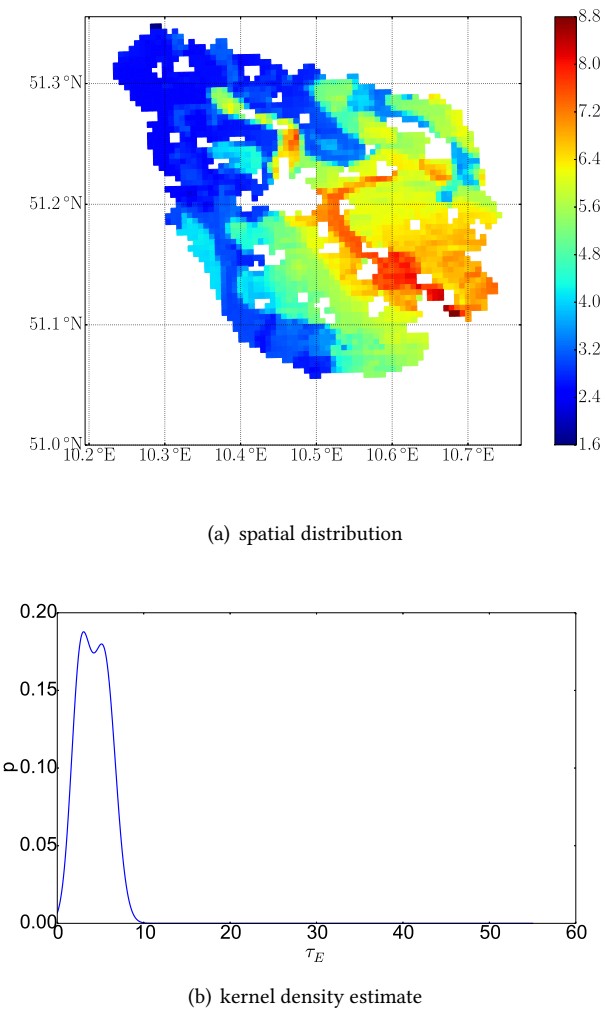

(a) spatial distribution

(b) kernel density estimate

**Figure 20.** Mean life expectancy of soil moisture in the Nägelstedt catchment caused by rainy months. Panel (a) shows the spatial distribution and Panel (b) shows the kernel density of mean life expectancy (in months).

5      Results showed strong differences in mean life expectancy during rainy months compared to the scenarios discussed above (compare Figure 20 with Figures 8 and 18). Compared to wet years, we saw even lower mean life expectancy. This can be explained by the strong impact of the rain on soil moisture leading to a flushing of the soil. We also saw a similarly small variance and a nearly unimodal distribution of mean travel-time values.

|                      | Pre   | PET   | SSM   |
| -------------------- | ----- | ----- | ----- |
| mean life expectancy | 0.736 | 0.221 | 0.857 |

**Table 4.** $R^2$ values for several predictors for mean travel time (as caused by rainy months).

|        | all years | dry years | wet years |
| ------ | --------- | --------- | --------- |
| $QI_f$ | 0.150     | 0.061     | 0.173     |
| $QI_s$ | 0.298     | 0.239     | 0.317     |
| $QB$   | 0.552     | 0.700     | 0.512     |

**Table 5.** Relative contribution of the different fluxes to runoff generation.

In addition to that, we saw differences for the statistical correlation of mean life expectancy with precipitation, potential evapotranspiration and saturated soil moisture (see Table 4). Compared to the standard travel-time behavior, precipitation was slightly less explanatory with mean life expectancy. This was caused by lower overall variation in precipitation values, due to constraining our analysis to large values therefore excluding low and medium range rain events. In contrast to that, $R^2$ values for PET and SSM increased.

### 3.6 Relevance of TTD's for hydrological inference

The above results demonstrated the impact of certain soil properties, as implemented in mHM, on mean travel times using the $R^2$ metric as a measure. In addition to that statistical analysis, their relationship can further be elucidated by analyzing Equation (2) or (3). Assuming for example a very simple linear relationship for both $Q$ and $ET$ with respect to $S$ we get for Equation (2) the following

$$\overrightarrow{p}_Q(T_E, t_{\mathrm{in}}) = \frac{\alpha_Q}{\theta(t_{\mathrm{in}})} \exp\left(-\alpha_{ET} T_E\right) \exp\left(-\alpha_Q T_E\right). \tag{9}$$

Equation (9) shows how under such simplified assumptions, the TTD of such a CV would follow an exponential distribution with its mean travel time being related to the recession constants $\alpha_Q$ and $\alpha_{ET}$. As shown above, such an exponential behavior is visible in the mean behavior (see Figure 7 right), whereas non-stationary TTD's show this exponential behavior to be superimposed by the event-based nature of the governing fluxes (see Figure 7 left).

In addition to these differences, we also saw different mean travel-time behavior for different regimes (see above). These differences can be explained by the actual implementation of $Q$ and $ET$ in mHM, which is generally non linear (see Section 2.2). In order to assess the different roles of each soil process on discharge generation, we calculated the relative contribution of each outflow mechanism for each regime. The data in Table 5 show how much of the water that entered the soil during a given time and left eventually as discharge was leaving as baseflow $QB$, slow interflow $QI_s$ or fast interflow $QI_f$. On

|  | all years | dry years | wet years |
|---|---|---|---|
| $R^2$ | 0.6059 | 0.6954 | 0.3619 |

**Table 6.** $R^2$ values for recharge vs. mean travel times for different regimes.

average, baseflow contributed the most to discharge with fast interflow having the smallest share. This overall distribution became stronger pronounced during dry years with baseflow taking the largest share of outflow generation and fast interflow becoming negligible. For wet years this trend is reversed, with water entering the soil during rainy months having an almost equal distribution. These different weighs show the relative impact and therefore the relative information content that travel-
time distributions could contain, i.e., travel times in dry years are mostly the results of the successive processes leading eventually to base flow (see Figure 3), whereas travel times during storm events contain information on all discharge processes combined.

To further elucidate the relationship between the resulting mean travel times and certain model parameters, we performed a regression analysis comparing the recession constant for recharge with the mean travel times for different regimes. Results
confirmed the relationship described above with mean travel times during dry years showing the strongest correlation (see Table 6).

Such a high interdependency between certain model parameters and data from different flow regimes is not unique for TTD's. Using discharge alone would reveal similar overall tendencies, e.g., discharge data from droughts is more informative for calibrating baseflow recession constants. What is, however, new is the additional information content, which is not
contained in discharge data alone. Not only can this improve calibration efforts, it allows the inference of additional system states. This is in particular relevant, but not confined to, the total amount of stored groundwater. Discharge data are not sensitive to, and therefore not informative for, groundwater levels, but only to its relative changes. TTD's on the other hand, strongly depend on the total amount of water stored in every CV. Using both data types for inference would therefore allow to provide reasonable estimates of this quantity. Similarly, the estimation of water in the root and vadose zone can be improved.
In addition *Birkel and Soulsby* (2015) highlight the temporal aspects of travel times on model calibration. They point out, how the sampling frequency of the time series should match the expected travel times of the underlying process. Our results above revealed different time scales for different hydrological regimes, that differed by almost an order of magnitude. Despite this heterogeneity, all travel times in our study remained in the range of months. Under such circumstances, a high resolution measurement campaign with daily or even hourly intervals would not be necessary.
Although the above explanations provide only a limited perspective on the relationship between TTD's and model parameters, it can be said that the strong interlink between the travel-time behavior and outflow generation indicates the high information content of the former with respect to the latter. As a result, travel-time distributions should be regarded as highly informative for the calibration of hydrological models. As mentioned in the Introduction, *McDonnell and Beven* (2014) have made the case for the usefulness of TTD's for the parametrization of such models. The above presentations provide empirical
support for this notion.

## 4   Conclusions

In this study, we investigated the spatially-distributed soil-moisture dynamics in the Nägelstedt catchment by virtue of travel-time distributions. The states and fluxes, needed for the derivation of the travel times, were numerically computed using the mesoscale Hydrological Model (mHM), which was calibrated against 55 years of discharge data as well as using detailed data on soil properties, land cover and precipitation. We performed a statistical analysis of mean travel times to describe the soil response decoupled from the event-driven impact of precipitation.

Comparing the derived mean travel times for several modeling scales (spanning over one order of magnitude), we did not see any significant difference in their distribution. This indicates a general soundness of the parametrization scheme of mHM used for the calculation of the states and fluxes on the different modeling scales. Our analysis shows that precipitation, saturated soil moisture and potential evapotranspiration are strong statistical predictors of mean travel time behavior. We also note that on average shorter mean travel times correspond to forested area and larger ones to crop/grassland, an observation that we linked to both correlations between forested and high-precipitation areas as well as the different water uptake mechanisms of trees vs. crop/grass.

We also investigated the travel-time behavior for different hydrological regimes, i.e., for dry and wet conditions (using soil moisture and precipitation as indicators). Our analysis revealed significantly different travel-time behavior for each of these regimes. Despite the strong heterogeneity of soil properties as well as (to a lesser extent) precipitation values, we could discriminate these regimes also in the resulting distribution of mean travel times.

Under dry conditions, we saw mean travel times having a pronounced bimodal distribution with long mean travel times and large variance. Such long travel times reveal the strong impact of baseflow on the generated outflow, whereas the large variance shows the variety of soil responses under dry conditions. Such conditions are therefore suited for inferring soil properties relating to baseflow generation. In addition, due to the large variance of soil responses, such conditions would allow to infer the spatial origin of solutes found in discharge streams. Such inferences are however, hampered by the long travel times involved. Not only are long time series needed, measurements must also being performed during such dry conditions.

Under wet conditions, we saw mean travel times having a unimodal distribution with shorter mean travel times and a smaller variance. This shorter travel times are caused by a larger influence of the slow and fast interflows on the total discharge behavior. As a result, TTD's derived under such conditions may be suited for inferring the parameters relating to these hydrological processes.

In case of rainy months, which overlap with wet conditions to a significant degree, we saw a similar distribution of travel times, but with even shorter mean values. This indicates a stronger impact of fast interflow on the total discharge behavior. Such information can therefore be valuable for improving the parametrization of the fast interflow related processes.

It is important to emphasize that our results have been derived with respect to a single hydrological model, i.e., mHM, only. As a result, we also need to critically assess the limitations of this approach and its impact on the reliability of our conclusions. First, mHM treats the hydrological storage in every compartment as fully mixed. In the absence of additional information,

we consequently assumed a uniform sampling scheme for the discharge generation from every mHM cell. This may have introduced errors in the age distribution of fluxes and therefore the travel-time behavior as discussed in Section 2.1. Due to the well established ability of mHM to take subgrid heterogeneity into account, we have confidence in the physical plausibility of the spatially explicit soil moisture states and fluxes. In the absence of, say solute data, we have, however, to consider these assumptions as tentative and open to revision. The other limitation of our approach stems form the computational nature of our study introducing a number of uncertainties. Like any hydrological model, mHM may suffer from three different sources of uncertainty; input uncertainty, structural uncertainty and parametric uncertainty. We would therefore like to assess their nature and potential impact on our results and conclusions. First, input uncertainty is referring to the uncertainties inherit in the forcing of the model, i.e., precipitation. Our results have shown the strong impact of precipitation on travel-time behavior. It would therefore stand to reason to expect a strong impact of any uncertainty from precipitation to propagate to the resulting travel-time behavior. However, we investigated mean behavior only, where time series from many months were averaged. We therefore consider possible contributions to our results to be minor. Next, structural uncertainty depend on the conceptual implementation of subsurface processes within mHM and our choices of different mHM compartments for our analysis. In Section 2.2, we discussed this issue by providing the rationals for, e.g., including the interflow components into our analysis. Finally, parameter analysis is probably the largest total source of uncertainty and several studies have recently investigated its impact on mHM output generation (*Samaniego et al.*, 2013; *Cuntz et al.*, 2015; *Livneh et al.*, 2015). The studies show that, while the fluxes are typically well represented in mHM (*Livneh et al.*, 2015), the overall soil moisture storage showed less accuracy, in particular during droughts *Samaniego et al.* (2013). For droughts, our results showed in general long travel-times and pronounced soil specific behavior with comparably lesser impact of precipitation. While we do not expect a major impact on the qualitative nature of these results, we should consider the quantitative aspect, i.e., the specific values for mean travel times to be inconclusive. In general, we consider the uncertainty stemming from the storage estimate to be the most relevant due to having both comparably lower accuracy and the strong impact on overall travel-time behavior demonstrated above. This is exacerbated since the water content relevant for outflow generation may not be the same as the one relevant for travel-time behavior. Immobile water due to, e.g., dead-end pores, affects the latter but not the former. It is, however, this connection between the total water content and the resulting travel-time behavior that makes the use of TTD's an important tool for a better calibration of hydrological models.

As an outlook, we can say that, having established a comprehensive description for the storage and release of water in the investigated catchment, the natural next step is the integration of reactive solute transport. As demonstrated by e.g., *Botter et al.* (2010), the concept of travel-time distributions can directly be adapted to account for the transport of both conservative and reactive solutes. This extension would facilitate to compare our predictions with the wealth of data that has been and continues to be collected within the AquaDiva center at the Hainich Critical Zone Exploratory (*Küsel et al.*, 2016). Thereby, we will be able to test our predictions by virtue of a large data set as well as initiate the collection of additional new data.

## Appendix A: Forward and backward formulation of travel times

Both the forward and backward formulations for TTD's can be derived from Equation (1) by additionally associating each term with its distribution, so

$$\frac{d}{dt}[S(t)p_S(T,t)] = J(t)p_J(T,t) - ET(t)p_{ET}(T,t) - Q(t)p_Q(T,t). \tag{A1}$$

Here $T$ is a placeholder for either the age or life expectancy of the water parcel. The total derivative in Equation (A1) can be reformulated using the material derivative, so

$$\left(\frac{\partial}{\partial t} + \frac{d}{dt}T\frac{\partial}{\partial T}\right)[S(t)p_S(t,T)] = J(t)p_J(t,T) - ET(t)p_{ET}(t,T) - Q(t)p_Q(t,T). \tag{A2}$$

Equation (A2) is the general PDE describing the time evolution of the age of the water in the CV. It is worthwhile to note that there is a significant inconsistency in the literature with respect to this equation. *Botter et al.* (2011) discuss the backward formulation of Equation (A2), while referring to it as the Master Equation (ME). This is certainly justified given that the ME is describing the time evolution of the PDF of any Markov process, i.e., a stochastic process that is local in (chronological) time. This condition is true for Equation (A2). In addition, Equation (A2) is not only local with respect to chronological time $t$ but also with respect to the travel time $T$. Interpreting $T$ as $x$, it becomes obvious that Equation (A2) is analogous to the much simpler Fokker-Planck equation or, since there is no 'diffusion', the even simpler advection-reaction equation. On the other side, *Porporato and Calabrese* (2015) are careful to trace this equation back to the seminal work of both *M'Kendrick* (1925) and *von Förster* (1959) in population dynamics. Consequently, they call this equation the McKendrick-von Förster (MKVF) equation.

One problem with Equations A1 and A2 is the lack of closure, i.e., $p_S(T,t)$, $p_J(T,t)$, $p_{ET}(T,t)$ and $p_Q(T,t)$ are different variables. The solution to this problem is the formulation and/or derivation of a dependency, i.e., closure, between the storage and the fluxes through

$$p_F(T,t) = \omega(T,t)p_S(T,t). \tag{A3}$$

This closure function $\omega(T,t)$ must follow some properties to ensure the normality of both $p_S(T,t)$ and $p_F(T,t)$, with the latter being the PDF of a flux, i.e., effective precipitation, evapotranspiration or discharge. These closure functions are called StorAge Selection (SAS) functions in the literature (*Rinaldo et al.*, 2015). Several different formulations exist with the one given above being based on the work of *Botter* (2012).

The shape of the SAS function is determining the preference of the fluxes, e.g., discharge, for several ages of the water stored in the CV. In the backward formulation a flat function would correspond to no preference with respect to age, a monotonously decreasing function would correspond to a preference for younger water and a monotonously increasing function would correspond to a preference for older water

*Acknowledgements.* This project was financially supported by the Deutsche Forschungsgemeinschaft via the Sonderforschungsbereich CRC 1076 AquaDiva. In addition, Matthias Zink was financed by HGF-EDA. We furthermore kindly acknowledge our data providers: the German Weather Service (DWD), the Joint Research Center of the European Commission, the European Environmental Agency, the Federal Institute for Geosciences and Natural Resources (BGR), the European Water Archive, the Global Runoff data Centre, the project
5    CarboEuropeIP (EU-FP6) as well as Axel Don for the provision of eddy covariance data. .

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
