# Peer review of "Spatially Distributed Characterization of Soil-Moisture Dynamics Using Travel Time Distributions"

_Hydrology and Earth System Sciences, 2016_

## Referee Comment (RC1) · H. McMillan (Referee) · 13 Jun 2016

This paper describes postprocessing of the state and flux data from the distributed hydrological model mHM, to calculate modelled transit time distributions of water in each model grid cell, in a catchment in Germany. The authors are then able to analyse the temporal and spatial variation in modelled transit time distributions, and relate this to variations in model inputs and parameters.

In its current form, the paper provides an investigation of mHM model behaviour that has not previously been studied, however there is no comparison to measured data to show whether or not the described model behaviour might be realistic, and as highlighted in my comments 2 and 3 below, there are aspects of the model which could be expected to disagree with field data. I therefore believe there is more work to do

to convince the reader of the scientific contribution of the paper, and why we should believe these model results.

In particular, I have the following major comments:

1. The introduction reads as though no previous studies have done what this paper sets out to do, i.e. to use the states and fluxes of a hydrological model to gain additional information about transit time behaviour that cannot be directly extracted from field measurements (also referred to as the "virtual experiment" approach). However, this is not the case, and previous studies have done exactly this including Hrachowitz et al (2013), Fenicia et al (2010), my own McMillan et al (2012), and Sayama et al (2009) who include an analysis of grid size sensitivity and effect of soil properties on transit time, both analysed here. The novel aspect of this paper is that a distributed rather than lumped model is used, and I suggest that the introduction is changed to make this point.

2. The method used in the paper implicitly assumes instantaneous total mixing of the water in each model grid cell, without any partial mixing behaviour or inactive storage component. However, this type of process description is widely considered to be not representative of field conditions, and typically does not give model estimates comparable with field data estimates of mean residence time (see McDonnell et al 2010 and the previous references I gave). The authors need to discuss this point and justify the use of model results that rely on this assumption.

3. The paper analyses transit time distributions within each model cell. Because the model then routes water between cells in a downstream direction, I infer that the larger the catchment, the greater the modelled transit time because water must pass through a series of cells. However, transit time is not generally found to have a clear relationship with catchment scale (e.g. Hrachowitz et al., 2009). Again, the authors need to discuss and justify this point.

References:

A new time-space accounting scheme to predict stream water residence time and hydrograph source components at the watershed scale. T Sayama, JJ McDonnell - Water Resources Research, 2009

Assessing the impact of mixing assumptions on the estimation of streamwater mean residence time. Fabrizio Fenicia, Sebastian Wrede, Dmitri Kavetski, Laurent Pfister, Lucien Hoffmann, Hubert H. G. Savenije and Jeffrey J. McDonnell. Hydrological processes, 2010.

How old is streamwater? Open questions in catchment transit time conceptualization, modelling and analysis. McDonnell et al. Hydrol. Process. 24, 1745-1754 (2010)

Catchment Transit Times and Landscape Controls-Does Scale Matter? Hrachowitz et al., Hydrological Processes 24(1):117 – 125, 2009

---

## Referee Comment (RC2) · Anonymous Referee #2 · 25 Jul 2016

This manuscript uses the theory of travel time distributions in time variant flow systems and a spatially distributed hydrological model (mHM) to analyze the spatial distribution of mean travel times and life expectancies in a German catchment. TO the best of my knowledge, this is one of the first attempts of using spatially explicit formulations to analyze the main physical controls on travel time distributions.

Overall, I'm in favor of publication of this manuscript in HESS. The topic is timely and interesting, the technical analysis of the authors is largely robust, and the paper is quite clear (event though some improvements in the presentation are recommended, see below).

In what follows I provide a list of general suggestions and comments that need to be considered before the acceptance of the manuscript.

Title: i'm wondering if "soil moisture dynamics" would be a better choice instead of "soil dynamics"

Page 5, section 2.2: I suggest adding more information about the rationale behind these equations, and the assumptions (e.g. random sampling)

page 6, line 10: maybe it is worth adding more info about the nature of these global parameters gamma

equation (5b): need to add "if x5 > TV"

page 7 line 28: better specify where the runoff data are gathered (only at the outlet)

page 8, lines 3-4: maybe it is worth to show the result mentioned here, or provide explicit reference about where these results can be found in the existing literature

Figure 5: units are missing

Equations (7), (2) and results: it has been shown that the storage involved in solute circulation is much bigger than the hydrological storage that can be estimated using a rainfall runoff model. Most of the existing tracer data suggest this instance in many place around the world (Plynlimon, Hubber Brook, etc). This would imply the use of a larger storage in the denominator of eq (7) for the calculation of TTD. While I think this issue can not be addressed in the absence of chemical data, I think it would be important to make a discussion on this point and clarify the assumptions underlying the analysis (i.e. absence of residual storage).

When you apply the formulation to the scale of a single grid cell, then you have to include the effect of input and output lateral fluxes. Maybe it is worth to specify how the TTDs are calculated in a spatially distributed setting.

Page 10, lines 2-4: gamma models for stationary TTD are much more widespread than exponential models in the literature. Moreover, the ref to rodriguez-Iturbe anfd Valdez can be misleading, as in that case only the IUH is concerned.

[Figure]

Page 10, lines 22-24. SAS functions have been introduced before Rinaldo et al., 2015 - only the name has been introduced later in those papers.

PAge 10, line 33: "the most simple SAS" should read "uniform SAS" with some references.

Page 11, lines 1-4: I would suggest to expand this discussion and provide more arguments / clarify your reasoning. Maybe the point here is that the mixing taking place at spatial scales smaller than 2 km X 2 km is not relevant?

Pages 13 -20: It would be good to see more discussion here about the physical interpretation of the results. this would increase significantly the breadth of the paper.

Page 20, equation not-numbered. Iì'm wondering why this equation is used to introdcude Figure 15 as the life expectancy in not involved (unless I'm missing something)

Figure 15: physical interpretation of these results? units are missing

Page 21, line 1 and Page 23, line 29: which is the underlying physical interpretation of these results?

Page 21, line 10: in a geenral cell the influx is just precipitation of this include also lateral fluxes?

Page 27, equation (8): I think there is an extra _Q before the "=" sign.

Page 28, last line: why does this happen?

Appendix A, equations (a1) and (a2): I guess the signs of the Q terms are wrong.

Appendix A, lines 14-17. MKVF equations are expressed in a different form, in which TTDs and output fluxes are grouped together. This "caveat" makes a huge difference: the explicit presence of the hydrological fluxes in eq. (a1) allows for the use hydrological models for TTD inferences - as done in this nice paper - and the coupled modeling of flow and transport at catchment scale, which is the next step foreseen by the authors

(page 29, lines 25-30).

---

## Author Comment (AC1) · 9 Aug 2016

We would like to begin by saying that we really appreciated the comments of the reviewer and we think we could much profit from them. We are sure that they helped us to significantly improve our manuscript. In the following, we present these comments as well as our point-by-point response to all of them. In addition, we added the revised version of the manuscript with changes tracked as a supplement.

This paper describes postprocessing of the state and flux data from the distributed hydrological model mHM, to calculate modelled transit time distributions of water in each model grid cell, in a catchment in Germany. The authors are then able to analyse the temporal and spatial variation in modelled transit time distributions, and relate this to variations in model inputs and parameters.

[Figure]

In its current form, the paper provides an investigation of mHM model behaviour that has not previously been studied, however there is no comparison to measured data to show whether or not the described model behaviour might be realistic, and as highlighted in my comments 2 and 3 below, there are aspects of the model which could be expected to disagree with field data. I therefore believe there is more work to do to convince the reader of the scientific contribution of the paper, and why we should believe these model results.

With respect to a perceived lack of realism in our model results, we would like to provide more information that may help to build confidence in their overall soundness. First, mHM has been successfully applied to a wide range of catchments in different parts of the world and demonstrated its ability to provide reliable predictions on the hydrological dynamics in said catchments. In addition, we can also provide support for the soundness of the predictions of mHM in the considered catchment. In Figure 1 and 2 we show two different results. Figure 1 shows the discharge data vs. our prediction for a dedicated time interval between 1970 - 1979, demonstrating the good performance of mHM in this regard. What is more important, however, is the ability of mHM to also predict data that has not been used for calibration. To demonstrate that, we used data from an eddy-covariance station within the modeled catchment. The comparison shows again a good fit between observation and model prediction (see Figure 2). These results should provide the necessary confidence for our model predictions. To better convey these points, we also amended the manuscript, where we now present and discuss both above results.

In particular, I have the following major comments:

1. The introduction reads as though no previous studies have done what this paper sets out to do, i.e. to use the states and fluxes of a hydrological model to gain additional information about transit time behavior that cannot be directly extracted from field measurements (also referred to as the "virtual experiment2 approach).

However, this is not the case, and previous studies have done exactly this including Hrachowitz2013, Fenicia2010, my own McMillan2012, and Sayama2009 who include an analysis of grid size sensitivity and effect of soil properties on transit time, both analyzed here. The novel aspect of this paper is that a distributed rather than lumped model is used, and I suggest that the introduction is changed to make this point. We appreciate the additional perspective on the topic of travel-time behavior. In the earlier version of the manuscript, we did not fully acknowledge this prior work and therefore created a biased impression of our contribution. In the revised version of the manuscript, we are now able to describe a more accurate picture of the present state-of-the-science and therefore better position our study vis-a-vis this prior work.

2. The method used in the paper implicitly assumes instantaneous total mixing of the water in each model grid cell, without any partial mixing behavior or inactive storage component. However, this type of process description is widely considered to be not representative of field conditions, and typically does not give model estimates comparable with field data estimates of mean residence time (see McDonnell et al. 2010 and the previous references I gave). The authors need to discuss this point and justify the use of model results that rely on this assumption. We fully acknowledge the limitations mentioned by the reviewer. The scheme by ?, which we use in our study, does indeed assume perfect mixing in every control volume. Although this limitation has later been amended by introducing a function describing age-dependent outflow generation, we did not use the improved scheme as described by Botter2011. This was done because this formalism is still lacking a method for deriving this age-dependency for a given catchment when using water fluxes, only. In the absence of say tracer data, we have no way of inferring the possible mixing in every single cell. Using the framework of Information Theory, one should use in such a situation the assumption with the least information content, which in this case is the uniform model. We

tried, however, to estimate the possible influence of this decision on our study. As pointed out by Harman2015, age-dependent (i.e. non-uniform) outflow generation is the result of several sources of both external and internal heterogeneity in the catchment. The aforementioned ability of mHM to account for such variability, may therefore make the assumption of complete mixing more justified. Since we are lacking actual data to verify this assumption, we tested its plausibility by determining the mean-travel time on different spatial resolutions. We hypothesized that the assumption of perfect mixing should be better meet on finer scales, where more heterogeneity is explicitly modeled. Any divergence from this assumption should therefore be visible when comparing mean-travel times between finer and coarser resolution. Our failure to find such differences adds plausibility that the assumption of complete mixing does not introduce major errors. In the absence of actual data, we have, however, to consider these inferences tentative and open to revision. In the revised version of the manuscript, we now acknowledge these points both in Section "Travel-time distributions for a single control volume" when introducing the equations for the derivation of the TTD's as well as in Section "Impact of modeling resolution".

3. The paper analyses transit time distributions within each model cell. Because the model then routes water between cells in a downstream direction, I infer that the larger the catchment, the greater the modeled transit time because water must pass through a series of cells. However, transit time is not generally found to have a clear relationship with catchment scale (e.g. Hrachowitz2010). Again, the authors need to discuss and justify this point. In mHM the water is not routed from cell to cell but leaves every cell and enters the river network. Once in the river network, the water moves very fast (order of days) compared to movement within each cell (order of months). As a result, there is only a very weak scale effect originating from the catchment itself, in particular for such a moderately sized catchment as used in the study. Given our results, we would anticipate

scale effects only for really large catchments.

Please also note the supplement to this comment:
http://www.hydrol-earth-syst-sci-discuss.net/hess-2016-232/hess-2016-232-AC1-supplement.pdf

—————————————————

[Figure]

[Figure]

**Fig. 1.**

RMSE = 12.44 mm mon$^{-1}$
BIAS = 5.78 mm mon$^{-1}$
CORR = 0.94

Evapotranspiration [mm mon$^{-1}$]

2004    2005    2006

**Fig. 2.**

---

## Author Comment (AC2) · 9 Aug 2016

We would again like to begin by saying that we really appreciated the comments of the reviewer and we think we could much profit from them. We are sure that they helped us to significantly improve our manuscript. In the following, we present these comments as well as our point-by-point response to all of them. In addition, we added the revised version of the manuscript with changes tracked as a supplement.

[Figure]

**Anonymous Referee 2**

This manuscript uses the theory of travel time distributions in time variant flow systems and a spatially distributed hydrological model (mHM) to analyze the spatial distribution of mean travel times and life expectancies in a German catchment. To the best of my knowledge, this is one of the first attempts of using spatially explicit formulations to analyze the main physical controls on travel time distributions.

Overall, I'm in favor of publication of this manuscript in HESS. The topic is timely and interesting, the technical analysis of the authors is largely robust, and the paper is quite clear (event though some improvements in the presentation are recommended, see below).

1. Title: I'm wondering if "soil moisture dynamics" would be a better choice instead of "soil dynamics" We agree with the Reviewer's assessment and changed the title accordingly.

2. Page 5, section 2.2: I suggest adding more information about the rationale behind these equations, and the assumptions (e.g. random sampling) We extended the discussion of these equations mainly by addressing their limitations. However, to keep their introduction concise, we refer for further information to the established sources.

3. page 6, line 10: maybe it is worth adding more info about the nature of these global parameters gamma

   We added more information to the paragraph to better highlight the role of these parameters for our investigation. See Section "Numerical model" in the revised version of the manuscript.

4. equation (5b): need to add "if $x_5 > TV$" We changed this as suggested.

[Figure]

5. page 7 line 28: better specify where the runoff data are gathered (only at the outlet) Yes, they are collected at the outlet. We clarify this in the revised version of the manuscript.

6. page 8, lines 3-4: maybe it is worth to show the result mentioned here, or provide explicit reference about where these results can be found in the existing literature In the revised manuscript, we now provide more data on the soundness of the calibration scheme. This includes a representative plot for the generated outflux as well as a comparison of the predicted AET and measured values for a measurement station in the catchment. This demonstrates the soundness of the calibration scheme with respect both to data that was used for calibration (discharge) and data that was purely the result of the model (AET). See also Section "Study area and model set-up" in the revised manuscript.

7. Figure 5: units are missing We added the units to the caption of the figure.

8. Equations (7), (2) and results: it has been shown that the storage involved in solute circulation is much bigger than the hydrological storage that can be estimated using a rainfall runoff model. Most of the existing tracer data suggest this instance in many place around the world (Plynlimon, Hubber Brook, etc). This would imply the use of a larger storage in the denominator of eq (7) for the calculation of TTD. While I think this issue can not be addressed in the absence of chemical data, I think it would be important to make a discussion on this point and clarify the assumptions underlying the analysis (i.e. absence of residual storage). We fully agree with the comment of the reviewer that hydrological models (e.g. mHM) are more concerned with fluxes that with states (i.e. storage) since that's what they are calibrated against and consequently, that's what they are sensitive for. As a result, the storage used in this models has a high degree of uncertainty. From the very beginning of our analysis, we therefore tried to minimize the influence of such possibly erroneous results from our model. This notion became even

more pressing when our results demonstrated the strong influence that the state variable, i.e. the storage, often has on travel-time behavior. One of the decisions we made, was to confine our analysis to soil moisture only (hence the title of the manuscript), since previous results indicate the overall ability of mHM to estimate soil water content. However, unless better estimates could be provided for the groundwater (which we are working on), where most of the water is stored, we completely excluding this compartment of the water cycle from our analysis. Due to the comments of the reviewer, we became aware that these points were not properly formulated in the original manuscript and better emphasize them now in the revised version (see Section "Numerical model"). In addition, we also consider the strong sensitivity of mean travel times on storage to be one of the main messages of our study. Contrary to discharge data, which is determined by the fluxes, mean travel times are sensitive to both fluxes and states. This opens the door for a more robust and informative calibration procedure, which we try to outline in Section "Relevance of TTDs for hydrological inference". To better convey this important notion, we revised this section accordingly.

9. When you apply the formulation to the scale of a single grid cell, then you have to include the effect of input and output lateral fluxes. Maybe it is worth to specify how the TTDs are calculated in a spatially distributed setting. In mHM subsurface lateral fluxes are assumed to be unimportant compared to the vertical fluxes. This is a modelling assumption that allows for much of the implementation of mHM but at the same time puts some limitations on its applicability. For our modelling this has two limitations: (i) surface lateral fluxes must be present within every grid cell which limits the application to grid sizes in excess of several hundreds of meters and (ii) groundwater states and fluxes are highly error prone due to large sugsurface lateral flux components in aquifers. As a result, we limited ourr analyses to grid sizes with $500$ m minimum and we also confined our analysis to soil water content, only.

10. Page 10, lines 2-4: gamma models for stationary TTD are much more widespread than exponential models in the literature. Moreover, the ref to Rodriguez1979 can be misleading, as in that case only the IUH is concerned. We agree with the Reviewer's assessment and amended the manuscript to better convey this notion.

11. Page 10, lines 22-24. SAS functions have been introduced before Rinaldo2015 only the name has been introduced later in those papers. We fully agree with the reviewer and changed the manuscript to reflect this fact.

12. Page 10, line 33: "the most simple SAS" should read "uniform SAS" with some references. We agree and changed this as suggested.

13. Page 11, lines 1-4: I would suggest to expand this discussion and provide more arguments/clarify your reasoning. Maybe the point here is that the mixing taking place at spatial scales smaller than 2 km X 2 km is not relevant? When we started out our study, we were aware of the limitation regarding the specification of an age-dependent outflow function. The parametrization of such a function would necessitate measurements against we could fit our model predictions. Since we did not have such data (yet), we decided to use the outflow function which assumes least knowledge, i.e. a uniform sampling without any age preference whatsoever. In the next step, we wanted to estimate the overall error that could result from such a decision. To estimate the possible influence of this decision, we reasoned that a scale-dependent bias in the estimation of travel-time behavior would indicate the existence and possible strength of such an error. This is due to the multi-scale nature of mHM, where subgrid heterogeneity is taking into account by virtue of an upscaling scheme. Using a smaller grid resolution would make this heterogeneity explicit and therefore reveal any possible unaccounted subscale influence. The lack of any scale effect in our results indicates that mHM is able to take this sub-scale heterogeneity into account within the investigated

scale range (which is roughly an order of magnitude). We are aware, that this
is only covering one possible source of age-dependent outflow behavior and that
other unresolved heterogeneity (at even smaller scales or due to other subsur-
face properties not accounted for in mHM) would influence the outflow generation
as well. We therefore regard our analysis as tentative. In the revised manuscript,
we now discuss and better highlight this reasoning.

14. Pages 13 -20: It would be good to see more discussion here about the physical
interpretation of the results. This would increase significantly the breadth of the
paper. We agree that a physical interpretation of these purely statistical analysis
should be provided whenever possible. We therefore extended the discussion
and provided physical reasoning for several of the effects observed. This includes
particularly precipitation, ET and land cover, which have demonstrated to have
a major effect on travel-time behavior (see also the respective sections in the
revised version of the manuscript).

15. Page 20, equation not-numbered. I'm wondering why this equation is used to
introduce Figure 15 as the life expectancy in not involved (unless I'm missing
something) The equation is now numbered. The connection to Figure 15 is such
that the scatter plot describes the relationship between mean life expectancy and
mean age. This is now better highlighted in the revised manuscript.

16. Figure 15: physical interpretation of these results? units are missing We were
not able to come up with a plausible physically-based explanation for this rela-
tionship. In particular, since the observed effect is quite minor. Consequently,
we only acknowledge that, whatever difference exists between forward and back-
ward formulation, this difference is not very strong. We did, however, add the
units to the figure.

17. Page 21, line 1 and Page 23, line 29: which is the underlying physical interpreta-
tion of these results? With respect to Page 21, Line 1: we did not come up with

anything other than hindsight reasoning. Therefore, we want to refrain from speculation. With respect to Page 23, line 29, i.e. the unimodal structure of the mean life expectancy: This observation was made for wet years. This means that the soil was largely saturated and acting more of a conduit for precipitation events without imposing any characteristics on its own. We revised the manuscript to better reflect this reasoning.

18. Page 21, line 10: in a general cell the influx is just precipitation of this include also lateral fluxes? As mHM is conceptualized to perform on mesoscale catchments, it is assumed that no lateral fluxes between grid cells do exist. Thus, only vertical fluxes are considered at a particular location, for which the only influx is precipitation. Only the surface routing accounts for lateral fluxes for estimating river runoff.

19. Page 27, equation (8): I think there is an extra Q before the "=" sign. We agree with the Reviewer and changed this as suggested.

20. Page 28, last line: why does this happen? In the revised version of the manuscript, we added a paragraph to the Section "Land cover properties", where we elaborate on the possible causal factors for this observation (see also our answer above).

21. Appendix A, equations (a1) and (a2): I guess the signs of the Q terms are wrong. That's true. We changed this in the revised version of the manuscript.

22. Appendix A, lines 14-17. MKVF equations are expressed in a different form, in which TTDs and output fluxes are grouped together. This "caveat" makes a huge difference: the explicit presence of the hydrological fluxes in e.q. (a1) allows for the use hydrological models for TTD inferences - as done in this nice paper - and the coupled modeling of flow and transport at catchment scale, which is the next step foreseen by the authors (page 29, lines 25-30). We welcome the overall

positive content of the comment above. We do, however, not see any question or criticism that could be directly addressed. Maybe this was not intended, though.

Please also note the supplement to this comment:
http://www.hydrol-earth-syst-sci-discuss.net/hess-2016-232/hess-2016-232-AC2-supplement.pdf

—————————————————

[Figure]

**Supplement:**

[revised manuscript text omitted]

---

## Author Response (AR1)

**Editor Decision**

Thank you for your detailed replies to the the comments of the 2 reviewers. As you have acknowledged, both reviewers raised a couple of interesting points, a few of which are of considerable importance for the underlying interpretation of your results. I very much appreciate that you invested some time to addressed most comments in-depths. Yet, I feel that some points need a bit more attention in order to clarify your approach for the reader and to avoid misinterpretations:

We appreciated the comments of the Editor and think we could benefit from them. We are sure that they helped us to significantly improve our manuscript. In the following, we present these comments as well as our point-by-point response to all of them.

1. relating to comments of both reviewers, it remains unclear *how* pQ is constructed in your study. More specifically: which pQ does the reader actually get to see? Is it the age distribution of an incoming signal J that leaves a grid cell through *all* exit routes (i.e. Q and ET)? or is it only the age distribution of water leaving the unsaturated zone (Q and ET?) as I seem to understand from your reply No.8 to reviewer 2? In case groundwater storage was considered, how was the hydrologically passive storage accounted for (see e.g. the well known Fogure 1 in Zuber, 1986, Journal of Hydrology 86) - this is highly relevant for the age distributions as it exacerbates the dichotomy between celerity and velocity (McDonnell and Beven, 2014)! In addition, as you showed, mHM uses several soil layers. Is pQ a direct reflection of the total outflow from *all* soil layers? A further point is that only 5 years of warm up period were used. As the subsequently used "mean" age can be considerably sensitive to long tails, what is the potential of your estimates being biased towards too young water? In other words, even with 50 years modelling, is it conceivable that we miss some minor volumes of very old water here? These points need to be made much clearer - in the methods but also in results and discussion sections (particularily in the figures!).

   This first comment contains several distinct sub-comments. To address them appropriately, we will answer them individually.

   (a) How is pQ determined.

   We are sorry for this misunderstanding. Due to the Editor's as well as the reviewers' comments, we are now aware that this crucial aspect was not properly explained in the earlier version of the mansucript. In the revised manuscript, we address these concerns by elaborating on the question what storages and fluxes were used for the analysis. It is now made clear in the Introduction, the Methods section as well as throughout the Results section (plus figure captions) that all our analysis was confined to the soil layer, only. In particular, we present the rationals for this decision both in the Introduction and further elaborate on its ramifications in the Methods section.

   (b) What about the soil layers.

   The Editor is correct in his interpretation: we estimated the age distribution (TDDs) of water leaving the unsaturated zone (Q and ET). The groundwater part is not accounted in the analysis (see our elaborations above).

   (c) Warm-up period too short.

   The problem of a warm-up period that might be too short is only relevant for backward TTDs. Most of the analysis in our study was performed with respect to forward TTDs. In this case a similar problem exists with respect to a period of time at the end of the estimation period that may have to be discarded. We determined the length of this period by using the partitioning function $\theta$ as a measure. This quantity describes the ratio of a water parcel that is entering the CV at a given time and eventually leaves as discharge. Adding this value with the ratio of the water parcel that leaves as ET should always add up to one. We could therefore add up these quantities to determine how much of the time series we could use. Our investigation showed that only close to the last two years needed to be discarded from the analysis. In the revised version of the manuscript, we acknowledge this notion (see beginning of the Results and Discussion section in the revised manuscript).

2. Not everybody may be familiar with the details of mHM and of how the parameters are obtained in that model. It may therefore be important to clarify much earlier in the manuscript,

how the soil moisture storage capacity is determined. Attentive readers will otherwise wonder if some of the storage is unaccounted for if the value is a "normal" calibration parameter (i.e. potential "passive" storage" that within you period of application does not become hydrologically active but may provide a mixing volume; see above and McDonnell and Beven, 2014).

With respect to the way the soil moisture storage is determined, we now address this issue early on in the manuscript (see the Introduction in the revised manuscript) and later we elaborate on it in the Methods section. We also expanded our explanation which parts of mHM are parameters determined during calibration and which are determined as model outputs (see the Methods section in the revised manuscript). In addition, we refer to Zacharias 2007, where important aspects of the calibration process are described in more detail. Finally, we discuss the difference between the water content relevant for outflow generation and for travel-time behavior in the Conclusion section of the revised mansucrit.

3. I am missing a clear research question and/or research hypothesis this manuscript is looking at. Please add that at the end of the introduction.

   We aggree that the manuscript in its previous form did not state concisly the main reserach questions and novel contributions of the study. From our current perspective, we would summarize the main questions as follows: (i) How are spatially distributed quantities, in particular land-cover, precipitation and soil type, impact travel-time behavior, (ii) how do different hydrological regimes impact travel-time behavior and (iii) what is the inter-connection between travel-time behavior and specific conceptualization of different hydrological processes and how may these connections be used for a better model calibration. These points are now better stated in the manuscript (see Introduction in the revised manuscript).

4. Note that figure captions should be stand-alone. In other words, the reader should be able to understand the figure by reading the captions alone. At the present the captions a very vague and uninformative. I would encourage you to be more specific and detailed. Only one example: what are the soil classes 9, 38, etc. in figure 14. what type of soils are we talking about?

   We heeded the advise of the Editor and strongly revised the figures caption in the manuscript to make them more self explanatory (see figure captions in the revised manuscript).

In general, I would be glad if you could invest some more time in developing a more detailed and clear description of your methods and a stronger discussion of the limitations of the chosen approach (e.g. what about parameter uncertainty? uncertainty in age distributions? etc. and the effect on the interpretation)

We thank the Editor for the for these suggestions. In the revised manuscript, we have now added more material to address these two main points. First, with regard to the methods, we have significantely expanded our description on how the states and fluxes were generated in mHM and how the TTDs were computed using these states and fluxes. This point was already addressed above. Second, we also added material to the Conclusions where we now ciritically assess the limitations of our approach. In particular, we discuss the different sources of uncertainty in travel-time estimation. These include uncertainties arising due to input data, simplified model structure and parametric uncertainty. With respect to input data, we refer to prior studies on the validety of the mHM input data. With respect to mHM model structure, we discuss the potential impact of our definition of soil moisture. However, to fully assess this aspect, a model-to-model comparison would be necessary. We are interested to pursue such a comparison with different candidate models being considered. The last point is parameteric uncertainty. Here we refer to the literature that exists on this topic with respect to mHM and discuss the main results from these studies and how they may relate to the estimation of TTDs. This last point is particularly interesting and a full study may be appropriate. These points are all discussed in the Conclusions section of the revised version of the manuscript.

[revised manuscript text omitted]

---

## Author Response (AR2)

**Editor Decision**

Dear authors,

thank you very much for addressing in detail the reviewer comments. Before I can accept the paper for publication, I would like to encourage the authors to have the manuscript checked by a native speaker due to a considerable number of spelling errors. In places the grammar also does not seem to be fully correct and some of the wording comes across as a bit awkward. In addition, please make sure to format the figures in an appropriate and complete way so that all axis are not only given symbols but also the associated units. the same is true for the legends (i.e. colour codes): although it is mentioned in the figure captions, make sure to provide units. Best regards, Markus Hrachowitz

We appreciated the positive review of the Editor. We employed the help of a native speaker to correct the spelling and straigthen the grammer. In addition, we revised all figures in order to inlude the dimensions of the respective quantities. We did not include units for the colorbars since most of them represent dimensionless quantities. The only exception are the spatially-distributed mean travel times. These pictures are, however, meant for providing a qualitative outline of the data (i.e. just to showcase the spatial arrangement of the mean travel times) and are always accompanied the actual graphs for the quantitative description of the data (where the units are now provided). Below, you can find the a marked-up version of the manuscript, where the relevant changes are highlighted. We hope this revisions meet your demands and help the publication process.

[revised manuscript text omitted]